

**SWAT Modeling of Water Quantity and Quality in the Tennessee River Basin:**
**Spatiotemporal Calibration and Validation**
Gangsheng Wang[1,2,*], Henriette I. Jager[1,2], Latha M. Baskaran[1], Tyler F. Baker[3], Craig C.
Brandt[4]
[1]Environmental Sciences Division, Oak Ridge National Laboratory, Oak Ridge, TN 37831 USA
[2]Climate Change Science Institute, Oak Ridge National Laboratory, Oak Ridge, TN 37831 USA
[3]Tennessee Valley Authority, Knoxville, TN 37902 USA
[4]Biosciences Division, Oak Ridge National Laboratory, Oak Ridge, TN 37831 USA
*Corresponding Author: **Gangsheng Wang**
Bldg 4500N, Room F129S, MS-6301
Oak Ridge National Laboratory
Oak Ridge, TN 37831-6301
wangg@ornl.gov
Notice: This manuscript has been authored by UT-Battelle, LLC under Contract No. DE-AC05-
00OR22725 with the US Department of Energy. The United States Government retains and the
publisher, by accepting the article for publication, acknowledges that the United States
Government retains a non-exclusive, paid-up, irrevocable, world-wide license to publish or
reproduce the published form of this manuscript, or allow others to do so, for United States
Government purposes. The Department of Energy will provide public access to these results of
federally sponsored research in accordance with the DOE Public Access Plan
(http://energy.gov/downloads/doe-public-access-plan).



**Abstract**
Model-data comparisons are always challenging, especially when working at a large spatial
scale and evaluating multiple response variables. We implemented the Soil and Water
Assessment Tool (SWAT) to simulate water quantity and quality for the Tennessee River Basin.
We developed three innovations to overcome hurdles associated with limited data for model
evaluation: 1) we implemented an auto-calibration approach to allow simultaneous calibration
against multiple responses, including intermediate response variables, 2) we identified empirical
spatiotemporal datasets to use in our comparison, and 3) we compared functional patterns in
landuse-nutrient relationships between SWAT and empirical data. Comparing monthly SWAT-
simulated runoff against USGS data produced satisfactory median Nash-Sutcliffe Efficiencies of
0.83 and 0.72 for calibration and validation periods, respectively. SWAT-simulated water quality
responses (sediment, TP, TN, and inorganic N) reproduced the seasonal patterns found in
LOADEST data. SWAT-simulated spatial TN loadings were significantly correlated with
empirical SPARROW estimates. The spatial correlation analyses indicated that SWAT-modeled
runoff was primarily controlled by precipitation; sedimentation was controlled by topography;
and $NO_3$ and soluble P were highly influenced by land management, particularly the proportion
of agricultural lands in a subbasin.

**Keywords**: model calibration, validation, reservoir, runoff, SWAT, Tennessee River, water
quality





## 1 Introduction

The Energy Independence and Security Act (EISA) of 2007 set a target for US production of over 36 billion gallons of renewable fuels annually by 2022 (EISA, 2007). Because agricultural development has historically been associated with impacts on water quality (Dodds and Oakes, 2008), converting the lands needed to meet EISA targets heightened concerns for the nation's rivers and lakes, as well as for downstream estuaries. The health of waters in the Tennessee River Basin (TRB) is of particular interest because this region supports one of the most biologically diverse river fauna in North America (Haag and Williams, 2014; Keck et al., 2014). Previous studies have shown higher nutrient and sediment loadings in non-forested, human influenced watersheds in the TRB (Scott et al., 2002). Evaluating changes in water quality associated with large-scale regional shifts in land-use and management requires process-based modeling of hydrology and nutrient dynamics (Wellen et al., 2015). Process-based models are favored whenever projections beyond historical conditions are needed because these models incorporate the processes leading to change and do not require extrapolation of statistical relationships beyond the range represented in the data.

Process-oriented models like the Soil & Water Assessment Tool (SWAT) (Arnold and Fohrer, 2005; Srinivasan et al., 1998) incorporate current understanding of linkages between watershed properties and water quality responses, but they are also difficult to calibrate (Wang and Chen, 2012). Although evaluation of multiple responses simulated by spatially-distributed process-based models over time and space is strongly encouraged (Cao et al., 2006; Wellen et al., 2015), such comprehensive evaluations are limited by the availability of spatial and long-term temporal data. This challenge is compounded for models applied at a regional scale because monitoring efforts tend to be local in scale and of short duration, especially for water quality (Hoos and



McMahon, 2009). As such, we see a role for empirical models in the calibration and validation
of regional-scale models.
Empirical models have previously been fitted to spatial and temporal nutrient loads in the US
(Saad et al., 2011). The monthly instream nutrient fluxes were estimated using LOADEST
(LOAD ESTimator) developed by the United States Geological Survey (USGS) (Runkel et al.,
2004; USGS, 2015). LOADEST assists in developing regression models for estimating nutrient
loads or fluxes over a user-defined time interval based on functions of streamflow, time, and
additional user-specified variables (Runkel et al., 2004). The SPARROW (SPAtially Reference
Regressions On Watershed attributes) is also a model developed by USGS that relates water
quality measurements to characteristics of watersheds (Hoos and McMahon, 2009; Saad et al.,
2011) to estimate nutrient loads/fluxes. Both models represent empirical relationships most
important during the historical period and smooth out the noise inherent in fine-resolution
temporal water-quality measurements.
When seeking regional surveys suitable for calibration, data may not be available for exactly
the outputs produced by the model. However, flexibility in assimilating data can be achieved by
comparing against intermediate or synthetic response variables. Several SWAT calibration tools
are available, e.g., SWAT-CUP 2012 (Abbaspour, 2014), the Auto-Calibration tool (Van
Griensven, 2005), and the R-SWAT-FME framework (Wu and Liu, 2014); however, these tools
do not include intermediate or synthetic response variables to compare against. This limitation
prevented us from calibrating SWAT using the final water quantity and quality responses if the
corresponding observations (or datasets) were not available.
This paper presents solutions to the aforementioned challenges, including fitting regional-
scale SWAT model when there is limited spatial and long-term water quality data available and





representation of reservoirs in a highly regulated watershed. We describe efforts to implement
SWAT modeling of water quantity and quality for the TRB including the configuration of 22
reservoirs. We incorporate the Shuffled Complex Evolution algorithm (Duan et al., 1992) into
SWAT2012 to enable auto-calibration of the model against multiple hydrologic (i.e., water
quantity) and water quality response variables (including intermediate and synthetic response
variables) at multiple sites. We calibrate and validate SWAT water quantity and quality against
empirically modeled datasets available from the USGS throughout the conterminous US. We
also used functional validation to compare primary drivers controlling runoff and water quality
in the process-based SWAT model and the empirical models. Functional validation goes beyond
adding a stamp of approval (i.e., validation), instead comparing relationships to understand
differences and guide future modeling or data collection efforts. The approach described here can
be applied in other regions of the US where the required empirical models have been developed.

**2 Materials and Methods**
**2.1 Study Area**
The Tennessee River Basin (TRB), a tributary basin of the Mississippi River Basin, is located
in the southeastern part of the United States (USGS, 2014b) (Fig. 1). There are significant
physiographic differences in the eastern and western portions of the basin (Price and Leigh,
2006). Forest cover is the dominant natural vegetation in the basin. In the western portion,
alluvial plains produced rich soils. The middle of the basin, which was historically covered by
bottomland forest and prairie, now supports high percentages of pasture and cropland. Eastward,
the geology becomes more mountainous and dominated by limestone with sandstone ridges. The
easternmost portion of the basin lies in the rugged Blue Ridge and Southern Appalachian





provinces with relatively poor soils (Price and Leigh, 2006). The TRB area has a subtropical
climate (warm, humid summers, mild winters) (Sagona, 2003). December through early May is
the major flood season (TVA, 2014). Since the 1930's, the TRB has been impounded by a series
of dams (reservoirs), most of which are managed by the Tennessee Valley Authority (TVA).
Main-stem Tennessee River dams are operated in "run-of-river" mode to support river navigation
and generate hydroelectric power. Dams on the tributaries function as storage impoundments and
are used primarily for flood control (TVA, 2014). Kentucky Dam is 35 km (22 mi) upstream
from Paducah, Kentucky, where the Tennessee River flows northwest into the Ohio River (Fig.

122  1).

**2.2 Watershed Delineation and Definition of SWAT Hydrologic Response Units**

SWAT (Version 2012/Revision 627) was used to model water quantity and water quality
(Arnold et al., 2012). The Digital Elevation Model (DEM) data (1-arc-second, c.a. 30 m) for
TRB was downloaded from the National Elevation Dataset
(http://nationalmap.gov/elevation.html). We conducted watershed delineation in ArcSWAT
(Winchell et al., 2013) based on (i) USGS-defined 8-digit Hydrologic Unit Codes (Jager et al.,
2015) (HUC8, Fig. 1), and (ii) major stream gages and reservoirs (Fig. 1). Watershed delineation
of the TRB using the DEM resulted in a drainage area of 106,124 km$^2$. Twenty-two (22)
reservoirs were included in the SWAT setup (Fig. 1). SWAT includes a reservoir module that
can represent these waterbodies in the watershed (Chen et al., 2015; Wang and Xia, 2010). The
reservoir outflow may be calculated by one of the four methods provided by SWAT (Arnold et
al., 2012): (i) average annual release rate for uncontrolled reservoir; (ii) measured monthly
outflow; (iii) simulated controlled outflow with target release; and (iv) measured daily outflow.




The last method (i.e., IRESCO = 3, measured daily outflow) was adopted in this study and TVA
provided daily reservoir outflow rates from 1985 to 2013.
Hydrological Response Units (HRUs) represent unique combinations of soil type, slope, and
land use or land cover. STATSGO soil map units (Soil Survey Staff, 1994) that comprised more
than 10% of a subbasin were retained. We discretized slope into four categories: <1%, 1–2%, 2–
5%, and >5%. We used the 2009 Cropland Data Layer (CDL-2009) (USDA-NASS, 2014) to
represent land use/land cover (Jager et al., 2015). Natural vegetation in TRB is dominated by
forest (59.4%) and grassland (11.7%). The major crops in TRB are hay (non-Alfalfa, 9.7%),
soybeans (1.7%), and corn (1.5%). We retained land-use classes that comprised more than 2%
area of a subbasin. This protocol created a total of 4,026 distinct HRUs in 55 subbasins.
**2.3 Meteorological Forcings**
We downloaded historical meteorological observation from DAYMET (Thornton et al., 1997)
estimated for the center of each HUC8 (Fig. 1) over the period 1980–2014 (35 years). Daily
meteorological variables include total precipitation (mm), maximum and minimum temperatures
(ºC), and solar radiation (MJ m$^{-2}$ d$^{-1}$). Two additional variables (wind speed and relative
humidity) were estimated by the SWAT model's climate generator (Gassman et al., 2007). The
mean annual precipitation (MAP) on HUC8 units ranged from 1129 to 1715 mm with an average
of 1433 mm during 1980-2014.
**2.4 Model Calibration**
Existing auto-calibration routines in the SWAT model are not designed to calibrate against
intermediate response variables (e.g., HUC8 runoff and $NO_3+NO_2$). For this effort, we
incorporated the Shuffled Complex Evolution (SCE) algorithm (Duan et al., 1992) into the
source code of SWAT2012 model to implement auto-calibration (Fig. 2). SCE is a stochastic





optimization algorithm that has been widely used in calibration of hydrological models including
SWAT (Wang and Xia, 2010; Zhang et al., 2009). We calibrated 39 parameters (Table 1)
governing the hydrologic (i.e., water quantity) and water quality processes in SWAT. The 39
parameters were selected based on the sensitivity analyses in previous studies (Abbaspour et al.,
2007; Baskaran et al., 2010; Bekele and Nicklow, 2007; Santhi et al., 2001; Wang et al., 2014;
Wu and Liu, 2012). Generally, these parameters were calibrated step by step. The hydrologic
parameters (No. 1–14) were first calibrated against hydrologic response variables (i.e., water
quantity variables, e.g., streamflow or runoff). The second step was to calibrate the water quality
parameters (No. 15–39) using water quality measurements (e.g., sediment, nitrogen, and/or
phosphorus), where a subset of the parameters might be calibrated depending on the response
variables as described below.
Fifteen types of calibrations with regard to various response variables (See Supplement Table
S1) were defined in our current auto-calibration tool. The first five types correspond to five
hydrologic response variables: daily streamflow, monthly streamflow, daily reservoir storage,
daily soil water content, and monthly runoff on subbasin or HUC8; the next five types include
monthly nutrient (sediment, nitrogen, phosphorus) fluxes (metric tons per month); and the last
five types refer to instream monthly nutrient concentration (mg/L). Other response variables
could also be defined and added to this calibration framework.
Criteria used to assess model performance include:
(1) Nash-Sutcliffe Efficiency (NSE, Eq. 1):
$$NSE = 1 - \frac{\sum_{i=1}^{n}\left[Y_{sim}(i) - Y_{obs}(i)\right]^2}{\sum_{i=1}^{n}\left[Y_{obs}(i) - \overline{Y}_{obs}\right]^2} \qquad (1)$$



where $Y_{obs}$ and $Y_{sim}$ are the observed and simulated data, respectively; $\overline{Y}_{obs}$ is the mean value of
observations; and $n$ and $i$ denote the number of data points and the $i$th data, respectively. $NSE$ is
less than or equal to 1 and may be negative (Moriasi et al., 2007).
(2) Percent Bias (PBIAS, Eq. 2):
$$PBIAS = \frac{\overline{Y}_{obs} - \overline{Y}_{sim}}{\overline{Y}_{obs}} \times 100\% \qquad (2)$$
where PBIAS (Moriasi et al., 2007) denotes the deviation of predicted mean value ($\overline{Y}_{sim}$) from
observed mean value ($\overline{Y}_{obs}$) as a percentage of $\overline{Y}_{obs}$.

According to Moriasi et al. (2007), model simulation is satisfactory if $NSE > 0.5$ and if

$|PBIAS| \leq 25\%$ for streamflow (runoff), $|PBIAS| \leq 55\%$ for sediment, and $|PBIAS| \leq 70\%$ for
nitrogen (N) and phosphorus (P).

The overall objective function is the weighted average of individual objective functions:

$$F = \sum_{j=1}^{k} \left( w_j \cdot f_j \right) \qquad (3)$$
where $F$ denotes the overall objective function of $k$ individual objective functions; $f_j$ is the $j$th
objective function that could be calculated as $NSE$ or $|PBIAS|$ of interested response variable;
and $w_j$ is the weighting factor for each $f_j$.
**2.5 Calibration and Validation of Runoff**
Because streamflow (discharge) at a station within the TRB is largely a measure of the outflow
from the upstream reservoir(s) and because observed reservoir outflow was used in this study,
we calibrated hydrologic parameters based on runoff (i.e., total water yield) instead of
streamflow. We used the USGS computed monthly runoff (1985–1995) in HUC8(USGS, 2014a)





as reference for SWAT calibration, with one year (1985) for model spin-up and 10 years (1986–
1995) for model calibration. Another 18 years (1996–2013) of data were used for model
validation. The USGS HUC8 runoff estimates were generated by combining historical flow data
at USGS stream gages and the corresponding drainage basin boundaries and hydrologic units
boundaries.(USGS, 2014a) In a previous study, this dataset was used to calibrate the Variable
Infiltration Capacity (VIC) model for the conterminous US (Oubeidillah et al., 2014). The
objective of our multi-site calibration of runoff was to calibrate hydrologic parameters (No. 1–14)
of the subbasins within each HUC8. For example, when we implemented calibration in terms of
the HUC8-06010102, the parameters in four subbasins (1, 4, 5 and 6) in this HUC8 were
calibrated. We calculated simulated HUC8 runoff as the area-weighted-average of runoff from
subbasins within each HUC8:
$$R_{HUC8} = \sum_{j=1}^{m} \left[ R_{sub}(j) \times Area_{sub}(j) / Area_{HUC8} \right] \qquad (4)$$
$$Area_{HUC8} = \sum_{j=1}^{m} \left[ Area_{sub}(j) \right] \qquad (5)$$
where $R_{HUC8}$ is the runoff in a HUC8; $R_{sub}(j)$ and $Area_{sub}(j)$ are the simulated runoff (mm) and
the area (km$^2$) in the $j$th subbasin; and $Area_{HUC8}$ is the total area of the HUC8 that includes $m$
subbasins. The $NSE$ of monthly HUC8 runoff was defined as the objective function in hydrologic
calibration.
**2.6 Calibration and Validation of Monthly Nutrient Fluxes**
Nutrient measurements are sparse in rivers of the TRB. We have attempted to collect in-situ
water quality monitoring data from over 6,000 USGS and EPA (Environmental Protection
Agency) stations within the TRB through the National Water Quality Monitoring Council





(NWQMC)'s online Water Quality Portal  (WQP) (NWQMC, 2015). However, we did not find
long-term water quality data that coincided with our model simulation period (i.e., after 1980's).
Therefore, we used the LOADEST (LOAD ESTimator) dataset (Runkel et al., 2004) as reference
to calibrate water quality parameters. The LOADEST dataset provided estimates of monthly
nutrient fluxes (1996–2006) at the Tennessee River near Paducah, KY (i.e., the outlet of TRB)
(USGS, 2015). We used three-year (1994–1996) of data for model spin-up and 10 years (1997–
2006) for model calibration. Another seven years (2007–2013) of data were used for model
validation. Four water quality variables were available from the LOADEST dataset: sediment,
total phosphorus (TP), total nitrogen (TN), and $NO_3+NO_2$. The hydrologic parameters (No. 1–14)
calibrated against runoff were fixed during the calibration of water quality parameters (No. 15–
39). The *NSE* of monthly water quality was defined as the objective function during SWAT
calibration. When multiple response variables (e.g., Sediment, TP, TN, $NO_3+NO_2$) were
considered in model calibration, we used Eq. (4) to calculate the overall objective function. In
addition, the spatial distribution of mean annual nutrient loadings estimated by the SPARROW
model (Hoos and McMahon, 2009) was employed as another dataset for model validation at the
HUC8 level. The mean annual loads (MAL) of nutrients at the HUC8 level were calculated as
the area-weighted average of the MALs at all subbasins within the HUC8.
**2.7 Spatial Correlation Analyses**
Understanding how water yield and nutrient loadings vary with watershed characteristics is
important for quantifying primary drivers controlling water quantity and quality and for
developing nutrient management policies (Hoos and McMahon, 2009). To this end, we
implemented spatial correlation analyses between response variables and watershed attributes.
For this study, two variables were considered highly correlated if the absolute value of



correlation coefficient ($|r|$) was greater than 0.6 and the correlation was significant (p-value <
0.05), and moderately correlated if $|r|$ was between 0.2 and 0.6 and the correlation was
significant.
Based on the 29-year (1985–2013) simulation results from the calibrated SWAT model, we
calculated the mean annual values of response variables including Runoff, RC (Runoff
Coefficient, i.e., the ratio of runoff to precipitation), Sediment, OrgP (organic phosphorus), SolP
(soluble P), MinP (mineral P attached to sediment), TP, TN, OrgN (organic N), and $NO_3$. We
first conducted spatial correlation analysis between these response variables at the subbasin level.
We implemented spatial correlation analysis between the response variables and the subbasin
attributes (explanatory variables): Precipitation (mm), Subbasin_Slope (subbasin slope, %),
Elevation_Drop (difference between highest and lowest elevations, m), and fractions of major
land-use    types    (Forest_Fraction,    Grassland_Fraction,    Hay_Fraction,    Crop_Fraction,
Shrubland_Fraction, Wetlands_Fraction, Water_Fraction, Developed_Fraction, see Supplement
Fig. S1).
**3 Results and Discussion**
In the sections below, we describe calibration and validation of different SWAT model
responses including runoff and water quality metrics.
**3.1 Runoff**
SWAT simulations of TRB runoff were implemented with regard to the period from 1985 to
2013. We divided the 29-year runoff dataset into three sub-datasets: (i) a 1-year spin-up period
(1985), (ii) a 10-year calibration period (1986–1995), and (iii) an 18-year validation period
(1996–2013). The spatial resolution was the 8-digit hydrologic units (HUC8s) throughout the
TRB.





Hydrologic parameters (No. 1–14 in Table 1) were calibrated by comparing simulated
monthly HUC8 runoff with the USGS dataset. As an example, Fig. S2 shows the comparison
between the SWAT-simulated monthly runoff (i.e., water yield, denoted by 'Sim') and USGS
runoff (denoted by 'Obs') in HUC8-06040006, which is the outlet HUC8 of TRB. The *NSE*
values for this HUC8 were 0.90 and 0.70 for model calibration and validation, respectively.
Values of *NSE* across the 32 HUC8s (Fig. 3a) ranged from 0.56 to 0.93 with 50% confidence
interval (CI) of 0.74–0.88 (median 0.83); the *PBIAS* values (Fig. 3b) were within a narrow range
(−7%–13%). The model also performed well over the validation period, although *NSE* was lower
than that during the calibration period, as one would expect. The median *NSE* was 0.72 with 50%
CI of 0.57–0.77; and the *PBIAS* values were within the satisfactory range, i.e., ±25%,(Moriasi et
al., 2007) except for two HUC8s (06010108 and 06010204). Regarding the whole dataset for the
combined calibration and validation periods (1986–2013), the median *NSE* was 0.79 (50% CI:
0.69–0.84) and all of the *PBIAS* values were within ±25% except for one HUC8 with a
marginally satisfactory *PBIAS* (−26%).
The SWAT-simulated mean annual runoff (MAR) in the two aforementioned HUC8s
(06010108 and 06010204) might be more reasonable than the USGS-estimated MAR. We
analyzed the mean annual precipitation (MAP) and MAR data from 1986–2013 and found that
the runoff in these two HUC8s might be underestimated in the USGS dataset to some degree
(See Fig. S3).
**3.2 Water Quality**
The SWAT simulation of water quality began with the year 1996 owing to data availability. The
20-year (1994–2013) water quality dataset was also divided into three sub-datasets: (i) a 3-year





spin-up period (1994–1996), (ii) a 10-year calibration period (1997–2006), and (iii) a 7-year
validation period (2007–2013).
The water quality parameters (No. 15–39 in Table 1) were calibrated against the LOADEST
dataset by taking into account multiple objectives, i.e., four response variables, including
sediment, TP, TN, and $NO_3+NO_2$. Calibration greatly improved the performance of the model,
particularly for sediment ($NSE = -100$ and 0.06 for pre- and post-calibration, respectively), TP
($NSE = -2.5$ and 0.44 for pre- and post-calibration, respectively), and TN ($NSE = 0.02$ and 0.38
for pre- and post-calibration, respectively) (See Supplement Table S2). The $NSE$ values for
model validation were not as good as the $NSE$ for calibration, but the $PBIAS$ values (Table S2)
were satisfactory except for $NO_3+NO_2$ (−157%). The squared correlation coefficients ($r^2$) for TN
and TP during both calibration and validation periods equaled or exceeded 0.4 whereas the $r^2$
values for sediment and inorganic N were less than 0.4 (Table S2). SWAT-simulated water
quality responses reproduced the seasonal patterns found in LOADEST data during both
calibration and validation periods (See Fig. S4).
We further conducted the water quality simulation for a longer period of time (1985–2013)
than the period for model calibration and calibration (1997–2013). The spatial distributions of
SWAT-simulated MALs (1986–2013) of TN and TP were comparable to the SPARROW
estimates. The spatial patterns of SWAT-simulated TN and TP at the subbasin level are shown in
Fig. 4 and other variables (runoff, RC, sediment, and $NO_3$) are shown in Fig. S5. The spatial
MALs of TN and TP from SWAT were compared with the SPARROW dataset (MALs from
1975–2004) at the HUC8 level (Fig. 5). The $PBIAS$ values (between SWAT and SPARROW) for
TN (Fig. 5a) at 26 out of 32 HUC8 units were within the range of ±70%, and the $PBIAS$ values at
three HUC8 were higher than 80%. The 50% CIs of MAL of TN were 2.5–6.7 kg N/ha and 4.7–





7.4 kg N/ha by SWAT and SPARROW, respectively. The SWAT-simulated MAL of TN across
the 32 HUC8 units was 5.5 kg N/ha, which was 12% lower than the TN loading (6.2 kg N/ha)
estimated by SPARROW.
As for phosphorus (Fig. 5b), the SWAT-simulated MAL of OrgP+SolP (organic P + soluble
P) was 48% lower than the SPARROW-modeled TP, while the SWAT-simulated MAL of TP
(organic P + soluble P + mineral P) was 50% higher than the SPARROW-modeled TP. This was
because mineral P contributed most (75.2%) to the TP yield and organic P contributed least
(8.5%) to TP.  The SWAT-simulated MAL of OrgP+MinP (organic P + mineral P, 0.93 kg P/ha)
was comparable to the SPARROW-estimated TP (0.88 kg P/ha).
The spatial patterns of TN from the two models (SWAT and SPARROW) were significantly
correlated with each other ($r$ = 0.54, p-value < 0.001). The spatial pattern of SPARROW-
estimated TP was not significantly correlated with SWAT-simulated TP, but moderately
correlated with SWAT-simulated OrgP+SolP ($r$ = 0.38, p-value = 0.03) and highly correlated
with SWAT-simulated TN ($r$ = 0.84, p-value < 0.001).
Different from the multi-site hydrologic calibration for each HUC8, water quality was
calibrated against data from one site owing to data availability. Notice that this site (Paducah,
KY) is located at the outlet of TRB. In addition, 10 out of 25 water quality parameters are basin-
wide parameters (Table 1, denoted by 'basins.bsn') that are spatially identical in SWAT.
Therefore, current water quality calibration could represent the overall water quality regime in
the watershed. In summary, the temporal comparison of water quality simulations between
SWAT and LOADEST and the spatial comparison between SWAT and SPARROW showed a
correspondence between process-based SWAT modeling results and those from empirically
modeled data in the TRB.





**3.3 Spatial Correlation between Response Variables**
Functional validation seeks to compare key functional relationships found in process-based
models with those in data. This approach goes beyond simple 'validation' or casting a stamp of
approval on a model to understand the reasons for any remaining differences. We found that
SWAT-simulated MALs of MinP (mineral P attached to sediment) and TP were highly
correlated with sediment, which confirms that sediment plays an important role in watershed
phosphorus dynamics (Fig 6a). The TN yield was highly correlated with $NO_3$. TN loadings were
dominated by $NO_3$, i.e., the fraction of TN that was $NO_3$ ranged from 37% to 99% with an
average of 80%. TP was not correlated with TN, but OrgP (organic P) was moderately correlated
with OrgN (organic N) and SolP (soluble P) was moderately correlated with $NO_3$, which implies
similarity between SolP and $NO_3$ dynamics and similarity between OrgP and OrgN dynamics in
SWAT (Neitsch et al., 2011). The SPARROW-estimated spatial patterns of TN and TP were
correlated with each other; however, the SWAT-simulated spatial distributions of TN and TP
were decoupled because the MinP component (attached to sediment) in SWAT and TN was
dominated by inorganic nitrogen.  Nutrient (Sediment, P and N) loadings were not significantly
correlated with runoff (Fig. 6a), suggesting that nutrient point-source and non-point sources and
other physical landscape variables (Hoos and McMahon, 2009) control variation in nutrient
loadings simulated by SWAT.
**3.4 Correlation between SWAT Response Variables and Subbasin Attributes**
The spatial correlation analyses showed that the response variables differed in their controlling
factors. Runoff was highly correlated with precipitation ($r = 0.68$) and moderately and positively
related to Forest_Fraction ($r = 0.36$). The runoff coefficient (RC) was moderately and positively
correlated with Elevation_Drop ($r = 0.32$) and Subbasin_Slope ($r = 0.31$) (Fig. 6b).





Sediment loadings were moderately and positively correlated with Elevation_Drop ($r = 0.47$),
which verifies that the representation of topography and topology in this mountainous region
drives sediment dynamics (Wellen et al., 2015). We did not find any significant correlation
between TP and the aforementioned subbasin attributes. However, OrgP (organic P) was highly
associated with Developed_Fraction ($r = 0.64$) that represented human activities in urban area
(Hoos and McMahon, 2009); SolP (soluble P) was moderately correlated with Hay_Fraction ($r =$
$0.43$) indicating the influence of agricultural fertilization; and MinP (mineral P) was moderately
correlated with Elevation_Drop ($r = 0.37$) that was the primary driver for sediment generation.
Organic N (OrgN) was moderately correlated with Wetlands_Fraction ($r = 0.27$) and
Shrubland_Fraction ($r = -0.30$). $NO_3$ was highly correlated with Hay_Fraction ($r = 0.63$) and
moderately correlated with Crop_Fraction ($r = 0.48$), mostly owing to the response of $NO_3$ yield
to agricultural fertilization. In addition, $NO_3$ showed a moderate negative correlation with
Forest_Fraction ($r = -0.54$), Subbasin_Slope ($r = -0.44$), and Elevation_Drop ($r = -0.34$). Note
that TRB subbasins with steeper slopes generally had more forest and less cropland. The primary
drivers controlling TN were the same as those for $NO_3$ as TN was dominated by $NO_3$.
**4 Summary**
Model-data comparisons are always challenging, especially when working at a large spatial
scale and evaluating multiple response variables. We developed three innovations to overcome
hurdles associated with limited data for model testing: 1) we implemented an auto-calibration
approach to allow simultaneous calibration against multiple responses, including intermediate
response variables, 2) we identified empirical modeled datasets interpolated in space and time to
use in our comparison, and 3) we compared functional patterns in landuse-nutrient relationships
between SWAT and empirical data. Using these innovations, we were able to successfully





implement a calibrated model for the river basin and to evaluate performance. The SWAT
calibration tool developed in this study can be accessed upon request via GitHub
(https://github.com/wanggangsheng/SWATopt.git).
In addition to quantitative performance evaluation, we also discerned what the most
important influences on SWAT responses were. Runoff was mainly controlled by precipitation;
runoff coefficient and sedimentation were controlled by topographic attributes; whereas $NO_3$ and
soluble P were highly influenced by land use types, particularly the croplands (hay and other
crops). This is likely because our management of these croplands included applying fertilizers
containing N and P. Patterns in phosphorus dynamics differed more between the empirical and
process-based model than patterns in nitrogen dynamics, suggesting an area for future
exploration.

**AUTHOR INFORMATION**
**Corresponding Author**
*Tel: +1 (865)576-6685. E-mail: wangg@ornl.gov.
**Notes**
The authors declare no competing financial interest.

**Acknowledgements**
This material is based upon work supported by the U.S. Department of Energy, Office of Energy
Efficiency & Renewable Energy's Bioenergy Technology Program. Oak Ridge National
Laboratory is managed by UT-Battelle, LLC for the U.S. Department of Energy under Contract
No. DE-AC05-00OR22725. We thank Michelle Thornton for providing DAYMET data and





David Gorelick and Jasmine Kreig for discussion on this study. In addition, we appreciate the
insightful reviews of Drs. Shih-Chieh Kao and Sujithkumar Surendran Nair.






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





**Tables**

Table 1. Selected SWAT parameters for model calibration

| No | Parameter[a] | Description | Default | Min | Max | Input file | Fortran code |
|---|---|---|---|---|---|---|---|
| 1 | CN2 | Initial SCS curve number II | 85 | 35 | 98 | *.mgt | Readmgt.f |
| 2 | ESCO | Soil evaporation compensation factor | 0.95 | 0.01 | 1 | *.hru | Readhru.f |
| 3 | EPCO | Plant uptake compensation factor | 1 | 0.01 | 1 | *.hru | Readhru.f |
| 4 | OV_N | Manning's n value for overland flow | 0.1 | 0.01 | 0.6 | *.hru | Readhru.f |
| 5 | CH_N2 | Manning's n value for main channel | 0.014 | 0.01 | 0.5 | *.rte | Readrte.f |
| 6 | CH_K2 | Channel effective hydraulic conductivity (mm/hr) | 0.001 | 0.001 | 150 | *.rte | Readrte.f |
| 7 | ALPHA_BF | Baseflow alpha factor (days) | 0.048 | 0.001 | 1 | *.gw | Readgw.f |
| 8 | GW_DELAY | Ground water delay (days) | 31 | 0.0001 | 500 | *.gw | Readgw.f |
| 9 | RCHRG_DP | Deep aquifer percolation fraction | 0.05 | 0.0001 | 1 | *.gw | Readgw.f |
| 10 | GW_REVAP | Groundwater revap coefficient | 0.02 | 0.02 | 0.2 | *.gw | Readgw.f |
| 11 | GW_SPYLD | Specific yield for shallow aquifer ($m^3/m^3$) | 0.003 | 0.0001 | 0.4 | *.gw | Readgw.f |
| 12 | SOL_AWC | Available water capacity (mm $H_2O$/mm soil) | 0.2 | 0.01 | 0.4 | *.sol | Readsol.f |
| 13 | SOL_K | Saturated hydraulic conductivity (mm/h) | 10 | 0.01 | 100 | *.sol | Readsol.f |
| 14 | SURLAG | Suface runoff lag coefficient (days) | 4 | 0.5 | 12 | sub.lag | Readhru.f |
| 15 | SPCON | Linear re-entrainment parameter | 0.0001 | 0.0001 | 0.01 | basins.bsn | Readbsn.f |
| 16 | SPEXP | Exponent re-entrainment parameter | 1 | 1 | 2 | basins.bsn | Readbsn.f |
| 17 | PRF | Adjustment factor for sediment routing in the main channel | 1 | 0.001 | 2 | basins.bsn | Readbsn.f |
| 18 | ADJ_PKR | Adjustment factor for sediment routing in tributary channels | 1 | 0.5 | 2 | basins.bsn | Readbsn.f |
| 19 | CH_COV | Channel cover factor | 0.001 | 0.001 | 1 | *.rte | Readrte.f |
| 20 | CH_EROD | Channel erodibility factor | 0.001 | 0.001 | 1 | *.rte | Readrte.f |
| 21 | USLE_K | Soil erodability factor | 0.28 | 0.01 | 0.65 | *.sol | readsol.f |
| 22 | BIOMIX | Biological mixing coefficiency | 0.2 | 0.01 | 1 | *.mgt | Readmgt.f |
| 23 | RSDCO | Residue decomposition factor | 0.05 | 0.02 | 0.1 | basins.bsn | Readbsn.f |
| 24 | NPERCO | Nitrogen percolation factor | 0.2 | 0.001 | 1 | basins.bsn | Readbsn.f |
| 25 | N_UPDIS | N uptake distribution parameter | 20 | 0.001 | 100 | basins.bsn | Readbsn.f |
| 26 | NSETLR | N settling rate in reservoir (m/yr), Line 7 & 8 | 5.5 | 1 | 15 | *.lwq | Readlwq.f |
| 27 | SHALLST_N | Concentration of $NO_3$ in groundwater (mg N/L) | 0.0001 | 0.0001 | 1000 | *.gw | Readgw.f |
| 28 | ERORGN | Organic N enrichment for sediment | 0.001 | 0.001 | 5 | *.hru | Readhru.f |
| 29 | SOL_ORGN | Initial organic N concentration (mg N $kg^{-1}$ soil) | 0.01 | 0.01 | 50 | *.chm | Readchm.f |
| 30 | SOL_NO3 | Initial $NO_3$ concentration in the soil layer (mg N $kg^{-1}$ soil) | 0.01 | 0.01 | 50 | *.chm | Readchm.f |
| 31 | PPERCO | Phosphorus percolation factor (10 $m^3$ $Mg^{-1}$) | 10 | 10 | 17.5 | basins.bsn | Readbsn.f |
| 32 | PHOSKD | Phosphorus soil partitioning coefficient ($m^3$ $Mg^{-1}$) | 175 | 100 | 200 | basins.bsn | Readbsn.f |
| 33 | PSP | P sorption coefficient | 0.4 | 0.01 | 0.7 | basins.bsn | Readbsn.f |
| 34 | PSETLR | P settling rate in reservoir (m/yr), Line 5 & 6 | 10 | 2 | 20 | *.lwq | Readlwq.f |
| 35 | BC4 | Rate const for mineralization of organic P to dissolved P (1/d) | 0.35 | 0.01 | 0.7 | *.swq | Readswq.f |
| 36 | RS5 | Organic P settling rate (1/d) | 0.05 | 0.001 | 0.1 | *.swq | Readswq.f |
| 37 | ERORGP | Organic P enrichment ratio with sediment loading | 0.001 | 0.001 | 5 | *.hru | readhru.f |
| 38 | SOL_ORGP | Initial organic P (mg P $kg^{-1}$ soil) | 0.01 | 0.01 | 50 | *.chm | Readchm.f |
| 39 | SOL_SOLP | Initial soluble P concentration in the soil layer (mg P $kg^{-1}$ soil) | 5 | 0.01 | 50 | *.chm | Readchm.f |

[a]Four groups of parameters: No. 1–14: Water quantity; No. 15–21: Sediment; No. 22–30: Nitrogen; No. 31–39:

Phosphorus.





**Figure Captions**

Figure 1. Fifty-five subbasins and 22 reservoirs of the Tennessee River Basin (TRB) in the Soil and Water Assessment Tool (SWAT). The mainstem Tennessee River runs from east to west, exiting the basin below Kentucky Dam.

Figure 2. Integrating the Shuffled Complex Evolution (SCE) algorithm into the Soil and Water Assessment Tool (SWAT) permitted calibration against intermediate response variables.

Figure 3. Model calibration of SWAT-modeled runoff by optimizing hydrologic parameters and validation. The distribution shows values for 32 HUC8 units (8-digit Hydrologic Unit Codes). Measures of model performance are (a) Nash-Sutcliffe Efficiency (NSE), (b) Percent Bias (PBIAS).

Figure 4. Spatial distribution of SWAT-simulated mean annual values at 55 subbasins: (a) TN yield (kg N/ha), (b) TP yield (kg P/ha).

Figure 5. Comparison of spatial distribution of TN and TP yield between SWAT simulation and SPARROW dataset at 32 HUC8 units. SWAT metrics: OrgP_SolP = OrgP (organic P) + SolP (soluble P); OrgP_MinP = OrgP + MinP (mineral P attached to sediment); SolP_MinP = SolP + MinP; and TP = OrgP + SolP + MinP.

Figure 6. Spatial correlation analysis (a) between response variables (mean values from 1996 to 2013), (b) between response variables and subbasin attributes. Larger circle denotes higher





correlation coefficient and only significant correlations (p-value < 0.05) are shown. Numbers in

(a) denote correlation coefficients. Response variables are: sediment yield (kg TSS/ha), organic

phosphorus yield (OrgP, kg P/ha), soluble P yield (SolP, kg P/ha), mineral P yield attached to

sediment (MinP, kg P/ha), total P yield (TP = OrgP + SolP + MinP, kg P/ha), total nitrogen yield

(TN, kg N/ha), organic N yield (OrgN, kg N/ha), nitrate yield (NO3, kg N/ha), runoff depth

(mm), and runoff coefficient (RC, ratio of runoff to precipitation).





**Figures**

**Figure 1**

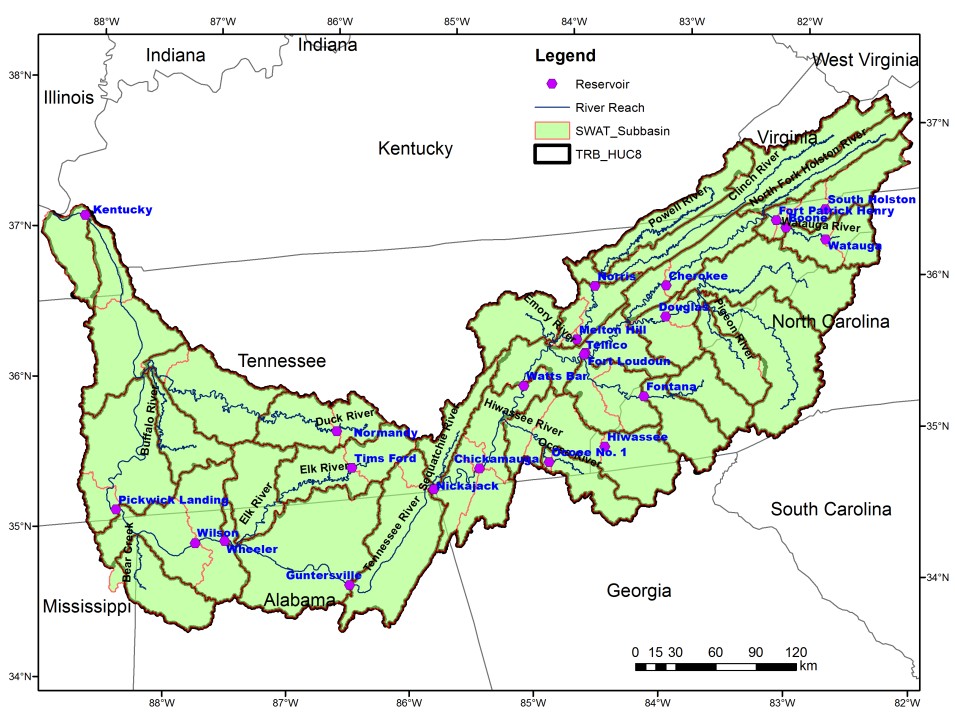





**Figure 2**

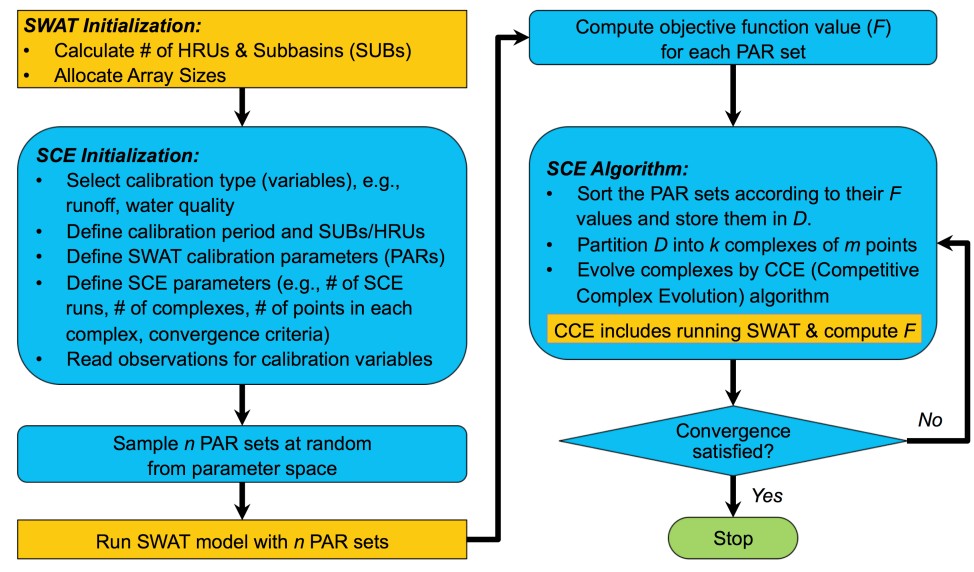




**Figure 3**

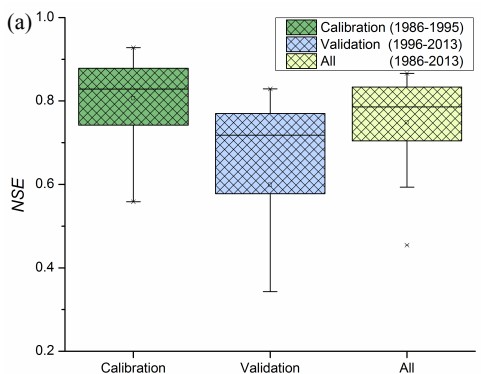
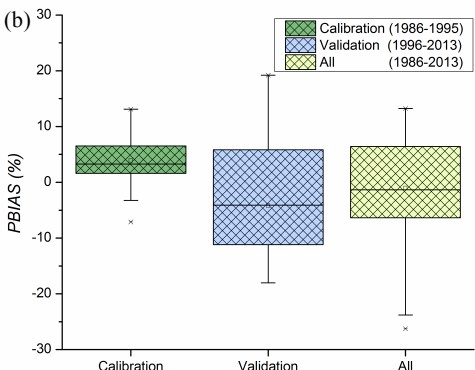



Figure 4

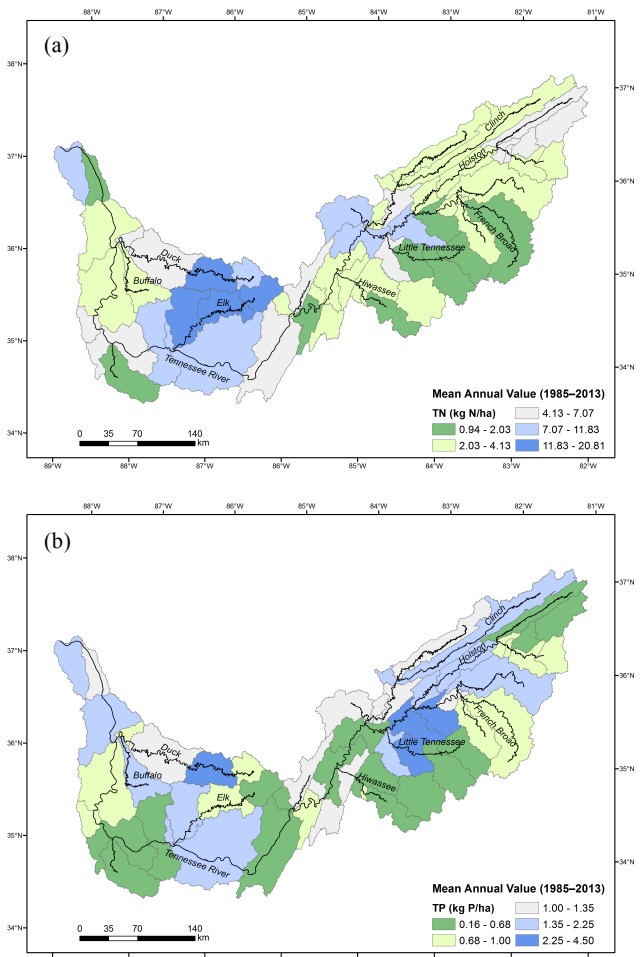




**Figure 5**

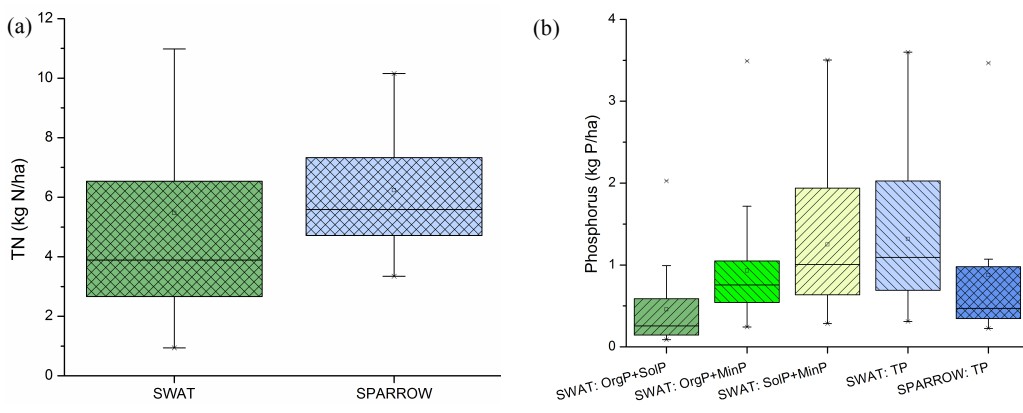



**Figure 6**

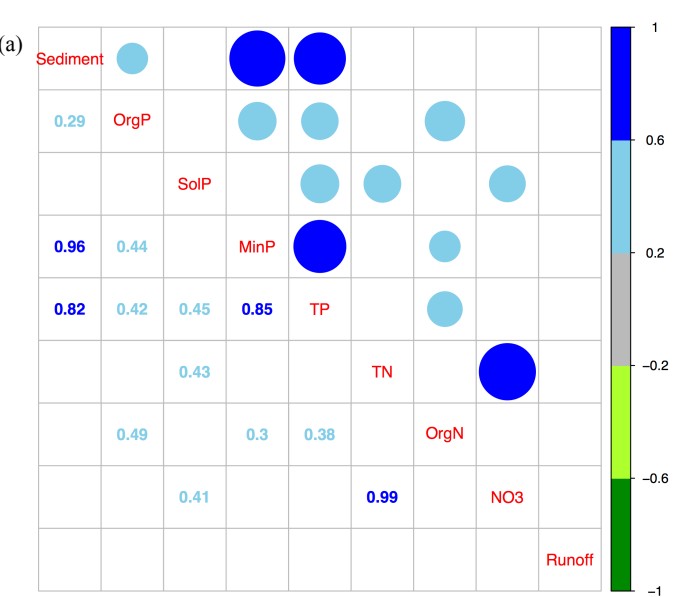

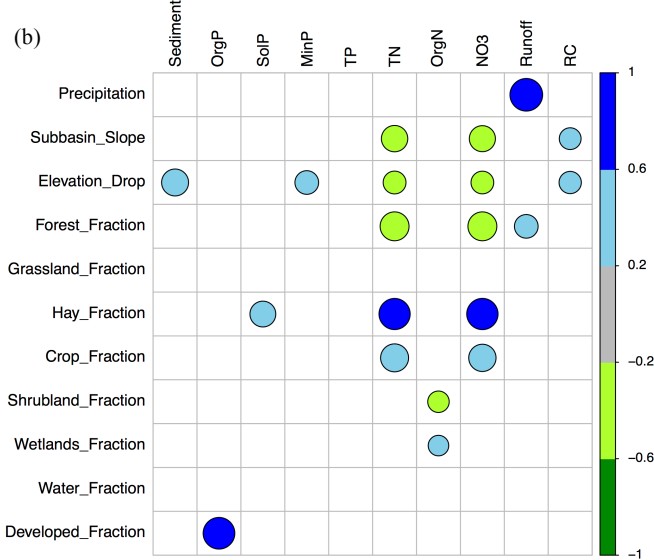