# Peer review of "SWAT Modeling of Water Quantity and Quality in the Tennessee River Basin"

_Hydrology and Earth System Sciences, 2016_

## Referee Comment (RC1) · Anonymous Referee #1 · 23 Feb 2016

**Short Summary:**

The manuscript outlines a framework for modeling water quantity and quality applying the Soil and Water Assessment tool (SWAT) to large river basins in the United States. The authors demonstrate their procedure with a case study in the Tennessee River Basin. To overcome shortcomings that are typical in such studies, for instance limitations in calibration data in terms of temporal and spatial resolution or the type of measured parameters, the authors state three innovations implemented in the study: 1) A parameter optimization framework was set up incorporating Shuffled Complex Evolution (SCE) as an optimization routine for SWAT that can provide intermediate response variables (variables that would not be recorded in the SWAT output files) for calibration.

[Figure]

2) For the calibration and validation as well as for model comparison, solely modeled data were incorporated, since no measured data were available within the considered time frame. 3) Correlating the results of several response variables to each other, and to the spatial properties of the modeled relationships among response variables as well between response variables and spatial attributes of SWAT sub basins.

**General Comments:**

With their work the authors raise important points eminent to almost every study analyzing in-stream water quality on large scale, such as limited data, particularly when it comes to nutrient fluxes, or identifying major contributors to nutrient loads. The study plots a workflow to incorporate the available datasets in the US in order to answer specific questions with process based modelling. When the approach is successful this might work as a blueprint for further studies on water quality in various US Catchments.

In general, I think the presented workflow of implementing available modelled data for analyzing US watersheds in a process based framework is a novel idea that could benefit further work on such topic. The outlined approach however, lacks some very critical considerations that must be covered before any publication of such framework. The in my opinion most critical points and my resulting suggestion to this manuscript are listed below:

1. My biggest concern is that the authors exclusively use modelled data derived from empirical models in their study! They state in the abstract, that they identified empirical data sets to compare model results to SWAT model responses. Further on in the paper however, the authors clearly state that the SWAT model was calibrated using computed runoff, as well as modelled LOADEST water quality data. These are clearly empirical datasets (which is mention in the manuscript further on and the authors are aware of). But most importantly, these datasets underlie large uncertainties in their responses. Therefore, implementing these

data, without thorough analysis of their quality is not an acceptable approach in my opinion. To prove the outlined approach as valid the calibrated (incorporating these empirical data) must be validated using observed data. The capabilities but also the shortcomings in comparison to observations require a detailed evaluation and discussion. The authors state that there is no observed data available for the model simulation period that can be used. This raises the question, which data was incorporated in the regression applying the LOADEST model in order to predict the nutrient loads for the respective time period. In a different paragraph the authors state that the major advantage of process based models over statistical models is the avoidance of extrapolation of statistical relationships. These statements contradict each other, when thinking of using extrapolated data from a statistical model to calibrate a process based model.

2. I do not agree with the authors statement, that the available automated calibration tools available for SWAT are incapable of including intermediate variables (as far as I understood this term right, these are variables comprised of other SWAT output variables, such as NO3 + NO2) in the calibration. When working with SWAT-CUP one can derive the product or the sum (etc.) of any variable that is evaluated in the subsequent post processing step. Therefore, less emphasis should be placed on this novelty in their study.

As the authors stated, these routines also offer SCE among other routines as sampling routine. Therefore, I do not agree with the authors that their routine is superior to others. I do however see the potential of this routine within the outlined workflow of specifically incorporating USGS data.

3. Apart from discharge, the performance of the SWAT model in its present state is low and the performance ratings are unsatisfactory for sediment, N, and P, especially when referring to Moriasi et al. (2007). As the authors used responses of a less complex model as calibration data for the more complex SWAT model, I question why SWAT is incapable of reproducing these results at all? The authors

state that SWAT is able to reconstruct the inter-annual repetitive patterns, but magnitudes of the peaks and periods with low nutrient loads are missed at all. Before publishing the model performance must improve. Any further model comparison of a poorly performing model to other models (in this case SPARROW) is in my opinion invalid.

4. Analyzing functional relationships of nutrient responses to any basin properties and their changes is in my opinion the actual strength of SWAT and therefore an interesting analysis to conduct. Many of the stated correlations however, are simply given by their functional relationship in the SWAT model and therefore obsolete to mention. The most surprising statement here was the very low correlation of sediment and P to runoff. These variables are usually highly correlated. Such findings require detailed analysis and discussion. The authors should not use SWAT as a black box, but should discuss the equations in the model in relationship to the simulated outputs. I advise the authors to restrict the thresholds of the correlation coefficient in their analysis. A correlation of 0.2 is not moderate in my opinion and gives the reader a wrong impression of the results. Taking the absolute value of the correlation coefficient leads to a loss of information.

Based on the given general statements I suggest **rejecting the paper** in its present form. I see however the potential of incorporating empirical data into a workflow of setting up SWAT models for US basins on HUC8 scale. Nevertheless, the above stated points must be conducted in order to prove this approach as valid.

**Specific Comments:**

*Figures:*

The figures do not fully express the results given in the manuscript, but rather provide a skewed view on them:

**Figure 3** only gives the results for runoff, but omits demonstrating any results for sed-
iment and nutrients. This gives a wrong impression of good results in general. The loads of N and P estimated using SPARROW are given as well on HUC8 scale. The authors only plotted the SWAT results for mean annual loads with spatial reference in figure 4. The comparison of the SWAT outputs to SPARROW are done in lumped form using box plots. A spatial comparison would be of much more interest, revealing which areas contribute which amounts of nutrient loads when using the two different models.

The results in **figure 5** require statistical testing. To me, it seems that there is a possibility that these two distributions are significantly different. In the text the authors say that the results are comparable, statistical testing can support or contradict this statement.

The abbreviation *"Obs"* in **figures S2** and **S4** is misleading. It is explained in the text how the term observation is used here, but only looking at the graphs the impression that the authors have used observed data is falsely conveyed.

*Paragraphs*

L 31: The term "intermediate response variable" requires explanation much earlier in the manuscript as it is not fully self-explaining but is a key term in this work.

L 66 – 69: I do not fully understand this statement.

L78 – 80: This sentence gives the impression that smoothing out information on nutrient loads is a benefit of empirical models. When using them for calibration of a process based model this might be a serious issue.

L85 – 89: As stated above I do not fully agree to this statement.

L91: The authors mention here that one challenge is that the watershed is highly regulated due to many reservoirs. This issue however, is not covered in the paper at all.

L140 – 141: What are the elevation range and topography in the catchment? How do the authors argue these thresholds for the slope classification?

L141 – 144: How did the authors assign or distribute the specific land uses in the

catchment?

L144: Corn might be one of the main contributors of P and N loads in the catchment, even if their fraction in the catchment is that low. What was the fraction of corn (or soybean) after applying a threshold of 2

L147 – 148: I would not mix up the terms estimations and observations and use them for the very same data.

L151: Which input was used for the weather generator?

L156: If the SWAT outlets are set at the same position as the location of HUC8 runoff data, then runoff is no intermediate variable.

L200: When such long time series are available I recommend to use a larger fraction of it as warm up period. One year of warm up is very short.

L209 – 210: Why is it necessary to accumulate these runoffs? If the sub basins in the model are assigned properly then the runoff or water yield is explicitly given by the SWAT outputs.

L218 – 222: This statement still surprises me. There are data from 6000 Stations but none of these are adequate for use in validation?

L244 – 246: As stated above the defined thresholds are too low. Taking the absolute refuses one to identify positive or negative relationships.

L272 – 277: For calibration the total range of the NSE values was given. For validation only the CI is shown. It is logical that some sub basins perform much worse during validation. Refusing to show this information gives however a wrong impression.

L294: I would not call it a great improvement when a model performs just as good as the mean value.

L321 -325: Why are SPARROW TP estimates highly correlated to SWAT TN estimates?

These originate usually from different processes. This finding requires further analysis and discussion.

**References:**

Moriasi, D. N., Arnold, J. G., Van Liew, M. W., Binger, R. L., Harmel, R. D., Veith, T. L. (2007). Model evaluation guidelines for systematic quantification of accuracy in watershed simulations. Transactions of the ASABE, 50(3), 885–900. http://doi.org/10.13031/2013.23153
* * *

---

## Referee Comment (RC2) · Anonymous Referee #2 · 1 Mar 2016

In this manuscript, the authors applied the SWAT model to the Tennessee River Basin to simulate water quantity and quality. Statistical modeling results were used to evaluate model performance. They found that model simulations were improved after parameter calibration. Correlation analyses were conducted to analyze the impacts of watershed attributes on water qualities. The authors have done lots of work in model simulation, calibration, and analysis. However, I think the manuscript needs to be substantially revised for publication. Here are my major concerns: First, I do not quite agree about the way how model performance evaluation is conducted. Although there are some difficulties in collecting observational data for model evaluation, field data should be the most valuable and reliable material for benchmarking. However, the

authors mainly used estimates from statistic models to calibrate and evaluate their modeling results. Since there are significant uncertainties in these statistic models, particularly LOADEST, comparing you model with these modeling results introduces additional uncertainties to calibration and validation of this work. As a result, the authors should include comparison with streamflow records, and concentrations of different elements from the USGS gauges in their work. Temporally explicit observations are limited, particularly for water quality variables, but the authors should at least compare the long-term averages. Second, although the authors claimed they did 'spatiotemporal 'calibration and validation in this work, I do not think this is well achieved. Only one gauge station was used, and I did not see any regional comparison maps between this study and SPARROW/regional runoff products. Instead, only regional means (figure 5) were compared and the spatial distribution of the selected variables from this work and the previous studies were not presented, which make it hard to validate results of this study. Third, spatial correlation analysis was not clearly introduced. I am wondering how the authors calculated the correlation coefficient? Did they use bivariate correlation or multiple linear regression? Did they consider the collinearity among the independent variables? Why only r was used to measure significance of the correlation, not P values? Finally, interpretation of the results, particularly the correlation analysis is insufficient. In addition to report significant correlations, the authors should explain the underlying mechanisms responsible for the correlation, and be cautious with non-causative correlations.

Specific comments:

Page 5, Line 97: but later you mentioned that only one site, close to the outlet of the basin, was used for model calibration Page 5, Line 106: you already provide the full name of this acronym on page 1. Page 7, Line 147: as far as I know daymet is a modeling dataset. Does it also provide original site level observation?

Table 1, it will be more helpful if you provide your calibrated parameter values, rather than providing the input file and fortran code.

[Figure]

Figure 6a is confusing. Consider to label variables in a different way.

---

## Author Comment (AC1) · 10 Mar 2016

Response to Interactive comment on "SWAT Modeling of Water Quantity and Quality in the Tennessee River Basin: Spatiotemporal Calibration and Validation" by G. Wang et al.

G. Wang et al. wangg@ornl.gov

[Response—] We would like to thank Referee #2 for his/her constructive comments. While you will be able to see the changes on the manuscript, we would like to high-light the following points: (1) We explained why we have to use empirical datasets (LOADEST and SPARROW), not the directly-observed datasets for model calibration

and validation. (2) We changed the thresholds of correlation between two variables and explained why there was very low correlation between sediment/P and runoff in our spatial analysis. (3) We added statistical tests to compare the TN and TP estimated by SWAT and SPARROW. We added Supplementary Fig. S6(a-d) to show spatial comparison (PBIAS, i.e., %bias) between SWAT and SPARROW modeled TN and TP. (4) We added Table S3 to show calibrated SWAT parameter values. Please see our point-by-point "Response to Comments" between [Response—] and [—Response] following the reviewer's original comments. [—Response]

Anonymous Referee #2

In this manuscript, the authors applied the SWAT model to the Tennessee River Basin to simulate water quantity and quality. Statistical modeling results were used to evaluate model performance. They found that model simulations were improved after parameter calibration. Correlation analyses were conducted to analyze the impacts of watershed attributes on water qualities. The authors have done lots of work in model simulation, calibration, and analysis. However, I think the manuscript needs to be substantially revised for publication. Here are my major concerns: First, I do not quite agree about the way how model performance evaluation is conducted. Although there are some difficulties in collecting observational data for model evaluation, field data should be the most valuable and reliable material for benchmarking. However, the authors mainly used estimates from statistic models to calibrate and evaluate their modeling results. Since there are significant uncertainties in these statistic models, particularly LOADEST, comparing you model with these modeling results introduces additional uncertainties to calibration and validation of this work. As a result, the authors should include comparison with streamflow records, and concentrations of different elements from the USGS gauges in their work. Temporally explicit observations are limited, particularly for water quality variables, but the authors should at least compare the long-term averages.

[Response—] Thanks for the reviewer's positive and thoughtful comments. We understand the reviewer's concerns on the calibration of hydrological model against empirical datasets, which differs from the traditional way to use observed data. At the beginning of this study, I also attempted to collect observations for SWAT calibration and validation. As mentioned in the manuscript, [Revised Manuscript Page 10 Line 216-220]" The utilization of streamflow (discharge) data for model calibration and validation in this study made little sense owing to the reservoir operation in the TRB. Because streamflow at a station within the TRB is largely a measure of the outflow from the upstream reservoir(s) and because observed reservoir outflow was used in this study, we calibrated hydrologic parameters based on runoff (i.e., total water yield) instead of streamflow." [Page 11-12 Line 242-253]" Nutrient measurements are sparse in rivers of the TRB. We have attempted to collect in-situ water quality monitoring data from over 6,000 USGS and EPA (Environmental Protection Agency) stations within the TRB through the National Water Quality Monitoring Council (NWQMC)'s online Water Quality Portal (WQP) (NWQMC, 2015). However, these observed data are not ready and useful for model calibration owing to the following reasons: (i) Although there are many measurement sites (stations), very few long-term time series are available within our study period (after 1980s); (ii) Not all of the water quality variables are measured at a specific site; and (iii) There is scaling issue regarding the water quality data. Due to limited sub-daily data points (i.e., one measurement in one month or several/many months), it is meaningless to do model calibration at daily scale. If we want to do model calibration at the monthly or yearly scale, we need to integrate the data from sub-daily to monthly or yearly scale, which is difficult when there are lots of data gaps." To our understanding, these are also the reasons why the LOADEST and SPARROW datasets were generated and they focused on temporal and spatial scale, respectively. The LOADEST dataset was time-series of monthly nutrient fluxes generated for a specific site, i.e., the Tennessee River near Paducah, KY, because this site had longer time-series of observation compared to the other sites; while the SPARROW dataset was the mean annual values of spatially-distributed nutrient loadings. These two published datasets were generated by the statistical approach and might underlie large

uncertainty, as pointed out the the reviewer. Thorough analysis of their quality might have been reported in relevant publications. However, we did not find such analysis pertaining to the TRB. We could do such analysis but we think it could be an independent study whereas it is not the focus of this manuscript. Without useful direct observations of water quality data, we have to use these two published datasets as reference to calibrate and validation the SWAT model for TRB. Through our preliminary communications with the USGS experts, they support the use of their empirical modeling as a way of getting the best of both worlds, empirical and process-based models. Also owing to the uncertainty in these empirical datasets, our calibration and validation performance was not satisfactory, which might NOT be regarded as unsuccessful. One could see that our SWAT simulations were capable of capturing the temporal patterns in the temporal LOADEST dataset and the spatial pattern of TN (total nitrogen) in SPARROW. The discrepancies in high or low values of water quality between SWAT and LOADEST resulted in the low model NSE values. Nevertheless, the mean values of mean annual loading (MAL) of SWAT-simulated TN and TP across the TRB were comparable to that of SPARROW estimated based on our statistical tests suggested by the reviewer: [Page 15 Line 335-337]: "The NSE values for model validation were not as good as the NSE for calibration, but the PBIAS values (Table S2) were satisfactory except for NO3+NO2 ($-157\%$)." [Page 16 Line 339-341]: "SWAT-simulated water quality responses reproduced the seasonal patterns found in LOADEST data during both calibration and validation periods (See Fig. S4)." [Page 16 Line 347-354]: "The LSD test indicated that the mean MALs of TN across the 32 HUC8 units were not significantly different between SWAT and SPARROW (p-value > 0.05) . . . . . . The PBIAS values (between SWAT and SPARROW) for TN at 26 out of 32 HUC8 units were within the range of $\pm70\%$, and the PBIAS values at three HUC8 were higher than 80% (Fig. S6a)." [Page 16-17 Line 355-365]: "The LSD test showed that the mean MAL of SWAT_TP (1.32 kg P/ha) was not significantly different from that of SPARROW_TP (0.88 kg P/ha) (Fig. 5b) . . . . . . The PBIAS values between SWAT_TP and SPARROW_TP at 13 out of 32 HUC8 units were within the range of $\pm70\%$ (Fig. S6b). In addition, the PBIAS

values between SWAT_OrgP+MinP and SPARROW_TP at 50% of the HUC8 units fell into the range of ±70% (Fig. S6c), and the PBIAS values between SWAT_OrgP+SolP and SPARROW_TP at 59% of the HUC8 units were within ±70% (Fig. S6d)" Our SWAT simulations also showed different, however more reasonable results for TP than SPARROW: [Page 18, Line 390-393]: "the SPARROW-estimated spatial patterns of TN and TP were correlated with each other; however, the SWAT-simulated spatial distributions of TN and TP were decoupled because MinP contributed most (65%) to TP and TN was dominated by inorganic nitrogen in SWAT." Overall, the observations in water quality were available but NOT ready and directly useful for model calibration and validation. The empirical temporal LOADEST and spatial SPARROW datasets were the best ones we could find at present to provide reference for our SWAT calibration and validation in the TRB. Our SWAT simulations could be an alternative to these datasets to characterize the water quality in the TRB, which is also the reason why we need SWAT modeling. Overall, the observations in water quality were available but NOT ready or directly useful for model calibration and validation. USGS are the experts on these data and that their synthetic data products reflect the best available use of observational data. The empirical temporal LOADEST and spatial SPARROW datasets were the best ones we could find at present to provide reference for our SWAT calibration and validation in the TRB. Our SWAT simulations could be an alternative to these datasets to characterize the water quality in the TRB, which is also the reason why we need process-based (e.g., SWAT) modeling. [—Response]

Second, although the authors claimed they did 'spatiotemporal 'calibration and validation in this work, I do not think this is well achieved. Only one gauge station was used, and I did not see any regional comparison maps between this study and SPARROW/regional runoff products. Instead, only regional means (figure 5) were compared and the spatial distribution of the selected variables from this work and the previous studies were not presented, which make it hard to validate results of this study.

[Response—] Thanks for the reviewer's constructive comments! We validated the nutrient yields using the spatial dataset SPARROW. We add Supplementary Fig. S6(a-d) to show spatial comparison (PBIAS, i.e., %bias) between SWAT and SPARROW modeled TN and TP. We added statistical tests to compare the TN and TP estimated by SWAT and SPARROW. [Page 13 Line 269-274] in "Materials and Methods": "We compared the TN and TP MALs between SWAT and SPARROW by (i) testing the significance of difference in mean MALs across TRB by the Fisher's least significant difference (LSD) method (De Mendiburu, 2015); (ii) testing the significance of difference in the probability distribution of the MALs by the Kruskal-Wallis (KW) test (Giraudoux, 2013); and (iii) calculating the PBIAS of MALs between SWAT and SPARROW at the HUC8 level. All statistical tests were conducted at the significance level of $\alpha$ = 0.05." [Page 16-17 Line 345-365] in "Results and Discussion": "The spatial distributions of SWAT-simulated MALs (1986–2013) of TN and TP were comparable to the SPARROW-estimated MALs (1975–2004) (Fig. 5 and Fig. S6). The LSD test indicated that the mean MALs of TN across the 32 HUC8 units were not significantly different between SWAT and SPARROW (p-value > 0.05), although the SWAT-simulated MAL (5.5 kg N/ha) was 12% lower than the SPARROW estimate (6.2 kg N/ha) (Fig. 5a). The 50% CIs of MAL of TN were 2.5–6.7 kg N/ha and 4.7–7.4 kg N/ha by SWAT and SPARROW, respectively (Fig. 5a). The KW test was significant, which implied that the MALs from the two models did not originate from the same probability distribution. The PBIAS values (between SWAT and SPARROW) for TN at 26 out of 32 HUC8 units were within the range of ±70%, and the PBIAS values at three HUC8 were higher than 80% (Fig. S6a). The SWAT-simulated TP (SWAT_TP) consisted of three components, i.e., organic P (OrgP), soluble P (SolP), and mineral P (MinP). The LSD test showed that the mean MAL of SWAT_TP (1.32 kg P/ha) was not significantly different from that of SPARROW_TP (0.88 kg P/ha) (Fig. 5b). The KW test indicated that the SPARROW_TP and SWAT_TP did not originate from the same probability distribution, but there was no evidence of stochastic dominance between SPARROW_TP and SWAT_OrgP+MinP or between SPARROW_TP and SWAT_OrgP+SolP (Fig. 5b). The PBIAS values between SWAT_TP and SPARROW_TP at 13 out of 32 HUC8 units were within the range of ±70% (Fig. S6b). In

addition, the PBIAS values between SWAT_OrgP+MinP and SPARROW_TP at 50% of the HUC8 units fell into the range of ±70% (Fig. S6c), and the PBIAS values between SWAT_OrgP+SolP and SPARROW_TP at 59% of the HUC8 units were within ±70% (Fig. S6d)" [—Response]

Third, spatial correlation analysis was not clearly introduced. I am wondering how the authors calculated the correlation coefficient? Did they use bivariate correlation or multiple linear regression? Did they consider the collinearity among the independent variables? Why only r was used to measure significance of the correlation, not P values?

[Response—] see [Page 13 Line 280-281]: "The bivariate correlation analyses were conducted using the 'cor' function in R (R Development Core Team, 2011) and the correlation were plotted by the 'corrplot' package (Wei, 2013)." Thus the collinearity among the variables was not considered in our study. Both P-value and correlation coefficient were used to measure significance of the correlation, only those correlations with p-value <0.05 were shown in Fig.6. please see Page 13 Line 282-285: "For this study, two variables were considered (i) highly correlated if the absolute value of correlation coefficient (|r|) was greater than 0.6 and p-value < 0.05, (ii) moderately correlated if |r| was between 0.4–0.6 and p-value < 0.05, and (iii) lowly correlated if |r| was between 0.2–0.4 and p-value < 0.05." [—Response]

Finally, interpretation of the results, particularly the correlation analysis is insufficient. In addition to report significant correlations, the authors should explain the underlying mechanisms responsible for the correlation, and be cautious with non-causative correlations.

[Response—] We agree with the reviewer's suggestions. Please see more explanations of underlying mechanisms following the presentation of the correlation, e.g., [Page 17-18 Line 382-403]: "We found that SWAT-simulated MALs of MinP (mineral P attached to sediment) and TP were highly correlated with sediment, which confirms

that sediment plays an important role in watershed phosphorus dynamics (Fig 6a). The TN yield was highly correlated with NO3. TN loadings were dominated by NO3, i.e., the fraction of TN that was NO3 ranged from 37% to 99% with an average of 80%. TP was not correlated with TN, but OrgP (organic P) was moderately correlated with OrgN (organic N) and SolP (soluble P) was moderately correlated with NO3, which implies similarity between SolP and NO3 dynamics and similarity between OrgP and OrgN dynamics in SWAT (Neitsch et al., 2011). In addition, the SPARROW-estimated spatial patterns of TN and TP were correlated with each other; however, the SWAT-simulated spatial distributions of TN and TP were decoupled because MinP contributed most (65%) to TP and TN was dominated by inorganic nitrogen in SWAT. Nutrient (Sediment, P and N) loadings were not significantly correlated with runoff in our spatial correlation analysis (Fig. 6a). This is because we were conducting spatial correlation analysis. If we implemented temporal correlation analysis for a specific HRU, subbasin or HUC8, taking the sediment MUSLE equation (Neitsch et al., 2011) as an example, the landscape factors (e.g., soil erodibility, land cover and management, support practice, topographic, coarse fragment) would not change or vary slightly with time, runoff would become the most important factor influencing sediment yield. Thus we could expect high correlations between sediment/P and runoff. The non-correlation between nutrient yields and runoff in our spatial analysis suggested that nutrient point-source and non-point sources and other physical landscape variables (e.g., topography and land cover) controlled spatial variation in SWAT-simulated nutrient loadings in the TRB." [Page 19 Line 409-410]: "Sediment loadings were moderately and positively correlated with Elevation_Drop (r = 0.47), which verifies that the representation of topography and topology in this region drives sediment dynamics (Wellen et al., 2015)." [Page 19 Line 412-416]: "OrgP (organic P) was highly associated with Developed_Fraction (r = 0.64) that represented human activities in urban area (Hoos and McMahon, 2009); SolP (soluble P) was moderately correlated with Hay_Fraction (r = 0.43) indicating the influence of agricultural fertilization; and MinP (mineral P) was lowly correlated with Elevation_Drop (r = 0.37) that was the primary driver for sediment generation." [Page

19 Line 418-423]: "NO3 was highly correlated with Hay_Fraction (r = 0.63) and moderately correlated with Crop_Fraction (r = 0.48), mostly owing to the response of NO3 yield to agricultural fertilization. In addition, NO3 showed a moderate and negative correlation with Forest_Fraction (r = −0.54) and Subbasin_Slope (r = −0.44). Note that TRB subbasins with steeper slopes generally had more forest and less cropland. The primary drivers controlling TN were the same as those for NO3 as TN was dominated by NO3." [—Response]

Specific comments: Page 5, Line 97: but later you mentioned that only one site, close to the outlet of the basin, was used for model calibration

[Response—] please see our response to the major concern #1. We calibrated runoff in 32 HUC8 units. We calibrated water quality in one site close to the outlet of TRB, but we validated TP and TN yields against the spatial dataset SPARROW. [—Response]

Page 5, Line 106: you already provide the full name of this acronym on page 1.

[Response—] We deleted '(TRB)' and kept the full name of 'Tennessee River Basin' because it is the lead sentence of this paragraph. [—Response]

Page 7, Line 147: as far as I know daymet is a modeling dataset. Does it also provide original site level observation?

[Response—] Yes, daymet is a modeling dataset. See [Page 8 Line 166-167]: "We downloaded synthetic meteorological data from DAYMET (Thornton et al., 1997) for the center of each HUC8 (Fig. 1) over the period 1980–2014 (35 years)" [—Response]

Table 1, it will be more helpful if you provide your calibrated parameter values, rather than providing the input file and fortran code.

[Response—] Thanks for the good suggestion! We added Table S3 to show calibrated SWAT parameter values. [—Response]

Figure 6a is confusing. Consider to label variables in a different way

[Response—] sorry for the confusing. This label style has been used in R packages to simplify the correlation plot. We added statement in the figure caption to further explain the figure, e.g., "Numbers in (a) denote correlation coefficients between the two variables shown in corresponding row and column." [—Response]

---

## Author Comment (AC2) · 10 Mar 2016

Response to Interactive comment on "SWAT Modeling of Water Quantity and Quality in the Tennessee River Basin: Spatiotemporal Calibration and Validation" by G. Wang et al.

G. Wang et al. wangg@ornl.gov

[Response—] We would like to thank Referee #1 for his/her constructive comments. While you will be able to see the changes on the manuscript, we would like to highlight the following points: (1) We explained why we have to use empirical datasets (LOAD-EST and SPARROW), not the directly-observed datasets for model calibration and val-

idation. (2) We used 'synthetic response variables' instead of 'intermediate response variables' in the revised manuscript. (3) We agree with the reviewer that available calibration tools also include synthetic variables. We regard our method as complementary to available calibration tools. We further explained why the HUC8 runoff is a synthetic variable and why it is necessary to calculate it based on SWAT output. (4) We changed the thresholds of correlation between two variables and explained why there was very low correlation between sediment/P and runoff in our spatial analysis. (5) We added statistical tests to compare the TN and TP estimated by SWAT and SPARROW. We added Supplementary Fig. S6(a-d) to show spatial comparison (PBIAS, i.e., %bias) between SWAT and SPARROW modeled TN and TP. (6) We added two more paragraphs to address the reservoir configurations and simulations in TRB. (7) We modified the notations in Figs.S2 and S4 to clarify the original 'Obs' data were empirical data. We also re-do Fig.6a-6b to show more classes of the correlation coefficient. Please see our point-by-point "Response to Comments" between [Response—] and [—Response] following the reviewer's original comments. [—Response]

Anonymous Referee #1 Short Summary: The manuscript outlines a framework for modeling water quantity and quality applying the Soil and Water Assessment tool (SWAT) to large river basins in the United States. The authors demonstrate their procedure with a case study in the Tennessee River Basin. To overcome shortcomings that are typical in such studies, for instance limitations in calibration data in terms of temporal and spatial resolution or the type of measured parameters, the authors state three innovations implemented in the study: 1) A parameter optimization framework was set up incorporating Shuffled Complex Evolution (SCE) as an optimization routine for SWAT that can provide intermediate response variables (variables that would not be recorded in the SWAT output files) for calibration. 2) For the calibration and validation as well as for model comparison, solely modeled data were incorporated, since no measured data were available within the considered time frame. 3) Correlating the results of several response variables to each other, and to the spatial properties of the modeled relationships among response variables as

well between response variables and spatial attributes of SWAT sub basins. General Comments: With their work the authors raise important points eminent to almost every study ana- lyzing in-stream water quality on large scale, such as limited data, particularly when it comes to nutrient fluxes, or identifying major contributors to nutrient loads. The study plots a workflow to incorporate the available datasets in the US in order to answer specific questions with process based modelling. When the approach is successful this might work as a blueprint for further studies on water quality in various US Catch- ments.

[Response—] Thanks for the reviewer's positive comments. [—Response]

In general, I think the presented workflow of implementing available modelled data for analyzing US watersheds in a process based framework is a novel idea that could benefit further work on such topic. The outlined approach however, lacks some very critical considerations that must be covered before any publication of such framework. The in my opinion most critical points and my resulting suggestion to this manuscript are listed below: 1. My biggest concern is that the authors exclusively use modelled data derived from empirical models in their study! They state in the abstract, that they iden- tified empirical data sets to compare model results to SWAT model responses. Further on in the paper however, the authors clearly state that the SWAT model was calibrated using computed runoff, as well as modelled LOADEST water qual- ity data. These are clearly empirical datasets (which is mention in the manuscript further on and the authors are aware of). But most importantly, these datasets underlie large uncertainties in their responses. Therefore, implementing these data, without thorough analysis of their quality is not an acceptable approach in my opinion. To prove the outlined approach as valid the calibrated (incorporating these empirical data) must be validated using observed data. The capabilities but also the shortcomings in comparison to observations require a detailed eval- uation and discussion. The authors state that there is no observed data available for the model simulation period that can be used. This raises the question, which data was incorporated in the regression applying the LOADEST model in order

to predict the nutrient loads for the respective time period. In a different paragraph the authors state that the major advantage of process based models over statis- tical models is the avoidance of extrapolation of statistical relationships. These statements contradict each other, when thinking of using extrapolated data from a statistical model to calibrate a process based model.

[Response—] Thanks for the reviewer's thoughtful comments. We understand the reviewer's concerns on the calibration of hydrological model against empirical datasets, which differs from the traditional way to use observed data. At the beginning of this study, I also attempted to collect observations for SWAT calibration and validation. As mentioned in the manuscript, [Revised Manuscript Page 11-12 Line 242-253]: " Nutrient measurements are sparse in rivers of the TRB. We have attempted to collect in-situ water quality monitoring data from over 6,000 USGS and EPA (Environmental Protection Agency) stations within the TRB through the National Water Quality Monitoring Council (NWQMC)'s online Water Quality Portal (WQP) (NWQMC, 2015). However, these observed data are not ready and useful for model calibration owing to the following reasons: (i) Although there are many measurement sites (stations), very few long-term time series are available within our study period (after 1980s); (ii) Not all of the water quality variables are measured at a specific site; and (iii) There is scaling issue regarding the water quality data. Due to limited sub-daily data points (i.e., one measurement in one month or several/many months), it is meaningless to do model calibration at daily scale. If we want to do model calibration at the monthly or yearly scale, we need to integrate the data from sub-daily to monthly or yearly scale, which is difficult when there are lots of data gaps." To our understanding, these are also the reasons why the LOADEST and SPARROW datasets were generated and they focused on temporal and spatial scale, respectively. The LOADEST dataset was time-series of monthly nutrient fluxes generated for a specific site, i.e., the Tennessee River near Paducah, KY, because this site had longer time-series of observation compared to the other sites; while the SPARROW dataset was the mean annual values of spatially-distributed nutrient loadings. These two published datasets were generated by the statistical approach and might underlie large uncertainty, as pointed out the the reviewer. Thorough analysis of their quality might have been reported in relevant publications. However, we did not find such analysis pertaining to the TRB. We could do such analysis but we think it could be an independent study whereas it is not the focus of this manuscript. Without useful direct observations of water quality data, we have to use these two published datasets as reference to calibrate and validation the SWAT model for TRB. Through our preliminary communications with the USGS experts, they support the use of their empirical modeling as a way of getting the best of both worlds, empirical and process-based models. Also owing to the uncertainty in these empirical datasets, our calibration and validation performance was not satisfactory, which might NOT be regarded as unsuccessful. One could see that our SWAT simulations were capable of capturing the temporal patterns in the temporal LOADEST dataset and the spatial pattern of TN (total nitrogen) in SPARROW. The discrepancies in high or low values of water quality between SWAT and LOADEST resulted in the low model NSE values. Nevertheless, the mean values of mean annual loading (MAL) of SWAT-simulated TN and TP across the TRB were comparable to that of SPARROW estimated based on our statistical tests suggested by the reviewer: [Page 15 Line 335-337]: "The NSE values for model validation were not as good as the NSE for calibration, but the PBIAS values (Table S2) were satisfactory except for NO3+NO2 ($-157\%$)." [Page 16 Line 339-341]: "SWAT-simulated water quality responses reproduced the seasonal patterns found in LOADEST data during both calibration and validation periods (See Fig. S4)." [Page 16 Line 347-354]: "The LSD test indicated that the mean MALs of TN across the 32 HUC8 units were not significantly different between SWAT and SPARROW (p-value > 0.05) . . . . . . The PBIAS values (between SWAT and SPARROW) for TN at 26 out of 32 HUC8 units were within the range of $\pm70\%$, and the PBIAS values at three HUC8 were higher than 80% (Fig. S6a)." [Page 16-17 Line 355-365]: "The LSD test showed that the mean MAL of SWAT_TP (1.32 kg P/ha) was not significantly different from that of SPARROW_TP (0.88 kg P/ha) (Fig. 5b) . . . . . . The PBIAS values between SWAT_TP and SPARROW_TP at 13 out of 32 HUC8 units were within the range of $\pm70\%$ (Fig.

S6b). In addition, the PBIAS values between SWAT_OrgP+MinP and SPARROW_TP at 50% of the HUC8 units fell into the range of ±70% (Fig. S6c), and the PBIAS values between SWAT_OrgP+SolP and SPARROW_TP at 59% of the HUC8 units were within ±70% (Fig. S6d)" Our SWAT simulations also showed different, however more reasonable results for TP than SPARROW [Page 18, Line 390-393]: "the SPARROW-estimated spatial patterns of TN and TP were correlated with each other; however, the SWAT-simulated spatial distributions of TN and TP were decoupled because MinP contributed most (65%) to TP and TN was dominated by inorganic nitrogen in SWAT." Overall, the observations in water quality were available but NOT ready or directly useful for model calibration and validation. USGS are the experts on these data and that their synthetic data products reflect the best available use of observational data. The empirical temporal LOADEST and spatial SPARROW datasets were the best ones we could find at present to provide reference for our SWAT calibration and validation in the TRB. Our SWAT simulations could be an alternative to these datasets to characterize the water quality in the TRB, which is also the reason why we need process-based (e.g., SWAT) modeling. [—Response]

2. I do not agree with the authors statement, that the available automated calibration tools available for SWAT are incapable of including intermediate variables (as far as I understood this term right, these are variables comprised of other SWAT output variables, such as NO3 + NO2) in the calibration. When working with SWAT-CUP one can derive the product or the sum (etc.) of any variable that is evaluated in the subsequent post processing step. Therefore, less emphasis should be placed on this novelty in their study. As the authors stated, these routines also offer SCE among other routines as sampling routine. Therefore, I do not agree with the authors that their routine is superior to others. I do however see the potential of this routine within the outlined workflow of specifically incorporating USGS data.

[Response—] Thanks for the reviewer's constructive comments. We agree that available calibration tools also include intermediate variables. for example, the Auto-

Calibration tool (Van Griensven, 2005) includes TKN, TN, and TP. SWAT-CUP can derive the product or the sum of different individual objective functions (e.g., squared error, R2, SSQR, PBIAS). We did not say our calibration routine is superior to others. We softened our statement and regarded our method as complementary to available calibration tools, see [Page 4 Line 81-87]: "Several SWAT calibration tools are available, e.g., SWAT-CUP (Abbaspour, 2014), the Auto-Calibration tool (Van Griensven, 2005), and the R-SWAT-FME framework (Wu and Liu, 2014), in which different objective functions in terms of multiple variables can be defined based on available observations for these response variables. Our SWAT simulation of TRB requires to calibrate the model against specific intermediate or synthetic response variables that have not been included in the aforementioned tools, such as the runoff depth at the HUC8 level and the total loading of NO3+NO2. " [—Response]

3.Apart from discharge, the performance of the SWAT model in its present state is low and the performance ratings are unsatisfactory for sediment, N, and P, espe- cially when referring to Moriasi et al. (2007). As the authors used responses of a less complex model as calibration data for the more complex SWAT model, I question why SWAT is incapable of reproducing these results at all? The authors state that SWAT is able to reconstruct the inter-annual repetitive patterns, but magnitudes of the peaks and periods with low nutrient loads are missed at all. Before publishing the model performance must improve. Any further model com- parison of a poorly performing model to other models (in this case SPARROW) is in my opinion invalid.

[Response—] Thanks for the reviewer's thoughtful comments. We understand the reviewer's concerns on the low model performance in calibration and validation of water quality in the TRB. Please see our response to the major concern #1, [—Response]

4.Analyzing functional relationships of nutrient responses to any basin properties and their changes is in my opinion the actual strength of SWAT and therefore an interesting analysis to conduct. Many of the stated correlations however, are simply given by their functional relationship in the SWAT model and therefore obsolete to mention. The most

surprising statement here was the very low corre- lation of sediment and P to runoff. These variables are usually highly correlated. Such findings require detailed analysis and discussion. The authors should not use SWAT as a black box, but should discuss the equations in the model in rela- tionship to the simulated outputs. I advise the authors to restrict the thresholds of the correlation coefficient in their analysis. A correlation of 0.2 is not moderate in my opinion and gives the reader a wrong impression of the results. Taking the absolute value of the correlation coefficient leads to a loss of information. Based on the given general statements I suggest rejecting the paper in its present form. I see however the potential of incorporating empirical data into a workflow of setting up SWAT models for US basins on HUC8 scale. Nevertheless, the above stated points must be conducted in order to prove this approach as valid.

[Response—] (1) Thresholds of correlation coefficient: We changed the thresholds of correlation between two variables, please see [Page 13 Line 282-285]: "For this study, two variables were considered (i) highly correlated if the absolute value of correlation coefficient ($|r|$) was greater than 0.6 and p-value $< 0.05$, (ii) moderately correlated if $|r|$ was between 0.4–0.6 and p-value $< 0.05$, and (iii) lowly correlated if $|r|$ was between 0.2–0.4 and p-value $< 0.05$. " (2) Positive or negative correlation: Actually we presented original (positive or negative) correlation coefficients in the text and Fig. 6. We only used the absolute value of the correlation coefficients to determine the high/moderate/low correlation. (3) The very low correlation between sediment/P and runoff: We added statements to explain the reason, See [Page 18 Line 393-403]: "Nutrient (Sediment, P and N) loadings were not significantly correlated with runoff in our spatial correlation analysis (Fig. 6a). This is because we were conducting spatial correlation analysis. If we implemented temporal correlation analysis for a specific HRU, subbasin or HUC8, taking the sediment MUSLE equation (Neitsch et al., 2011) as an example, the landscape factors (e.g., soil erodibility, land cover and management, support practice, topographic, coarse fragment) would not change or vary slightly with time, runoff would become the most important factor influencing sediment yield. Thus we could expect high correlations between sediment/P and runoff. The non-correlation
between nutrient yields and runoff in our spatial analysis suggested that nutrient point-source and non-point sources and other physical landscape variables (e.g., topography and land cover) controlled spatial variation in SWAT-simulated nutrient loadings in the TRB." [—Response]

Specific Comments: Figures: The figures do not fully express the results given in the manuscript, but rather provide a skewed view on them: Figure 3 only gives the results for runoff, but omits demonstrating any results for sed- iment and nutrients. This gives a wrong impression of good results in general. The loads of N and P estimated using SPARROW are given as well on HUC8 scale. The authors only plotted the SWAT results for mean annual loads with spatial reference in figure 4. The comparison of the SWAT outputs to SPARROW are done in lumped form using box plots. A spatial comparison would be of much more interest, revealing which areas contribute which amounts of nutrient loads when using the two different models.

[Response—] We presented calibration and validation results in Supplementary Table S2. We can move it from Supplement to the text if you like. We add Supplementary Fig. S6(a-d) to show spatial comparison (PBIAS, i.e., %bias) between SWAT and SPARROW modeled TN and TP. [—Response]

The results in figure 5 require statistical testing. To me, it seems that there is a possibility that these two distributions are significantly different. In the text the authors say that the results are comparable, statistical testing can support or contradict this statement.

[Response—] Thanks for the constructive comments! We added statistical tests to compare the TN and TP estimated by SWAT and SPARROW. See [Page 13 Line 269-274] in "Materials and Methods": "We compared the TN and TP MALs between SWAT and SPARROW by (i) testing the significance of difference in mean MALs across TRB by the Fisher's least significant difference (LSD) method (De Mendiburu, 2015); (ii) testing the significance of difference in the probability distribution of the MALs by the Kruskal-Wallis (KW) test (Giraudoux, 2013); and (iii) calculating the PBIAS of MALs

between SWAT and SPARROW at the HUC8 level. All statistical tests were conducted at the significance level of $\alpha = 0.05$." See [Page 16-17 Line 345-365] in "Results and Discussion": "The spatial distributions of SWAT-simulated MALs (1986–2013) of TN and TP were comparable to the SPARROW-estimated MALs (1975–2004) (Fig. 5 and Fig. S6). The LSD test indicated that the mean MALs of TN across the 32 HUC8 units were not significantly different between SWAT and SPARROW (p-value > 0.05), although the SWAT-simulated MAL (5.5 kg N/ha) was 12% lower than the SPARROW estimate (6.2 kg N/ha) (Fig. 5a). The 50% CIs of MAL of TN were 2.5–6.7 kg N/ha and 4.7–7.4 kg N/ha by SWAT and SPARROW, respectively (Fig. 5a). The KW test was significant, which implied that the MALs from the two models did not originate from the same probability distribution. The PBIAS values (between SWAT and SPARROW) for TN at 26 out of 32 HUC8 units were within the range of ±70%, and the PBIAS values at three HUC8 were higher than 80% (Fig. S6a). The SWAT-simulated TP (SWAT_TP) consisted of three components, i.e., organic P (OrgP), soluble P (SolP), and mineral P (MinP). The LSD test showed that the mean MAL of SWAT_TP (1.32 kg P/ha) was not significantly different from that of SPARROW_TP (0.88 kg P/ha) (Fig. 5b). The KW test indicated that the SPARROW_TP and SWAT_TP did not originate from the same probability distribution, but there was no evidence of stochastic dominance between SPARROW_TP and SWAT_OrgP+MinP or between SPARROW_TP and SWAT_OrgP+SolP (Fig. 5b). The PBIAS values between SWAT_TP and SPARROW_TP at 13 out of 32 HUC8 units were within the range of ±70% (Fig. S6b). In addition, the PBIAS values between SWAT_OrgP+MinP and SPARROW_TP at 50% of the HUC8 units fell into the range of ±70% (Fig. S6c), and the PBIAS values between SWAT_OrgP+SolP and SPARROW_TP at 59% of the HUC8 units were within ±70% (Fig. S6d)" [—Response]

The abbreviation "Obs" in figures S2 and S4 is misleading. It is explained in the text how the term observation is used here, but only looking at the graphs the impression that the authors have used observed data is falsely conveyed.

[Response—] Thanks for the reviewer's good suggestion! Fig.S2: We change "Obs"

to "USGS" and indicate that "USGS" represents "USGS Computed" in the caption Fig. S4: We change "Obs" to "LOADEST" We also use "SWAT" instead of "Sim" in Figs.2 and 4 to represent "SWAT-simulated". [—Response]

Paragraphs L 31: The term "intermediate response variable" requires explanation much earlier in the manuscript as it is not fully self-explaining but is a key term in this work.

[Response—] Thanks for the reviewer's good suggestion. We add explanation to "synthetic response variable", see [Page 1 Line 30-32]: "we implemented an auto-calibration approach to allow simultaneous calibration against multiple responses, including synthetic response variables, such as the HUC8 runoff as an area-weighted average of runoff in multiple subbasins;" [—Response]

L 66 – 69: I do not fully understand this statement.

[Response—] We deleted the confusing statement in terms of "local" and "regional" scales. Please see revised statement in [Page 3 Line 64-68]: "Although evaluation of multiple responses simulated by spatially-distributed process-based models over time and space is strongly encouraged (Cao et al., 2006; Wellen et al., 2015), such comprehensive evaluations are limited by the availability of spatial and long-term temporal data, especially for water quality (Hoos and McMahon, 2009). As such, we see a role for empirical data in the calibration and validation of the SWAT model for the TRB." [—Response]

L78 – 80: This sentence gives the impression that smoothing out information on nutrient loads is a benefit of empirical models. When using them for calibration of a process based model this might be a serious issue.

[Response—] Thanks for the comments. We delete the misleading statement and it now reads [page 4 Line 77-78]: "Both models represent empirical relationships most important during the historical period." [—Response]

L85 – 89: As stated above I do not fully agree to this statement.

[Response—] We revised this statement. See Page 4 Line 81-87: "Several SWAT calibration tools are available, e.g., SWAT-CUP (Abbaspour, 2014), the Auto-Calibration tool (Van Griensven, 2005), and the R-SWAT-FME framework (Wu and Liu, 2014), in which different objective functions in terms of multiple variables can be defined based on available observations for these response variables. Our SWAT simulation of TRB requires to calibrate the model against specific intermediate or synthetic response variables that have not been included in the aforementioned tools, such as the runoff depth at the HUC8 level and the total loading of NO3+NO2. " [—Response]

L91: The authors mention here that one challenge is that the watershed is highly regulated due to many reservoirs. This issue however, is not covered in the paper at all.

[Response—] We added two more paragraphs to address this issue, please see [Page 6-7 Line 131-151]: "SWAT includes a reservoir module that can represent waterbodies in the watershed (Chen et al., 2015; Wang and Xia, 2010). Twenty-two (22) reservoirs in the TRB were included in the SWAT setup (Fig. 1). We selected reservoirs based on the following guidelines: (i) The reservoir can be placed on a river reach generated by ArcSWAT's subbasin delineation (Winchell et al., 2013); (ii) The reservoir operation is managed by the Tennessee Valley Authority (TVA); and (iii) The reservoir is large, generally with storage capacity greater than 100,000 ac-ft (123 km3). Two reservoirs (Fort Patrick Henry and Ocoee No. 1) with storage capacity less than 123 km3 were also included because they are located near the outlet of HUC8-06010102 and HUC8-006020003, respectively. We collected the following reservoir variables (Arnold et al., 2012) as inputs to the SWAT model: year and month (denoted by SWAT variables IYRES and MORES) that the reservoir became operational; beginning and ending month of flood season (IFLOD1R and IFLOD2R); surface area (RES_ESA, ha) and volume (RES_EVOL, 104 m3) of water needed to fill the reservoir to the emergency spillway; surface area (RES_PSA, ha) and volume (RES_PVOL, 104 m3) of water needed to fill the reservoir to the principle spillway; and initial reservoir volume (RES_VOL, 104 m3) at the beginning of the simulation period. The reservoir outflow may be calculated by one of the four methods provided by SWAT (Arnold et al., 2012): (i) average annual release rate for uncontrolled reservoir; (ii) measured monthly outflow; (iii) simulated controlled outflow with target release; and (iv) measured daily outflow. The last method (i.e., IRESCO = 3, measured daily outflow) was adopted in this study and TVA provided daily reservoir outflow rates from 1985 to 2013. " [— Response]

L140 – 141: What are the elevation range and topography in the catchment? How do the authors argue these thresholds for the slope classification?

[Response—] [Page 6 Line 124-127] "The Digital Elevation Model (DEM) data (1-arc-second, c.a. 30 m) for TRB was downloaded from the National Elevation Dataset (http://nationalmap.gov/elevation.html), which indicates the elevation range of TRB from 56 to 2037 m." We classified the slope into 0-1, 1-2, 2-5, >5° because [Page 7 Line 155-159] "This analysis is part of a larger effort to model the entire Mississippi River Basin to understand influences on hypoxia in the Gulf of Mexico. Standardized slope classes were chosen to allow comparison of alternative tile drainage scenarios at <1% and <2% slope.Âǎ In addition, we require all HRUs with slope >5% to be included in the analysis because these steep areas are subject to erosion (Jager et al., 2015)." [—Response]

L141 – 144: How did the authors assign or distribute the specific land uses in the catchment?

[Response—] We used the USDA Crop Data Layer (CDL) 2009 as input, and used the ArcSWAT Land Use/Soils/Slope Definition tool to overlap the three data layers (land use/soils/slope) to generate HRU. [—Response]

L144: Corn might be one of the main contributors of P and N loads in the catchment, even if their fraction in the catchment is that low. What was the fraction of corn (or soybean) after applying a threshold of 2

[Response—] after applying a threshold of 2%, the fractions of corn and soybean are 1.5% and 1.7%, respectively. Actually, the previous statement showed the area fraction after applying the threshold. The original land use map showed fractions of 2.6% and 2.9% for corn and soybean, respectively. Our analysis showed that hay occupies 9.7% of TRB and [Page 20 Line 437-439] "NO3 and soluble P were highly influenced by land use types, particularly the croplands (hay and other crops)" [—Response]

L147 – 148: I would not mix up the terms estimations and observations and use them for the very same data.

[Response—] revised. See revised manuscript [Page 8 Line 166-167]: "We downloaded synthetic meteorological data from DAYMET (Thornton et al., 1997) for the center of each HUC8 (Fig. 1) over the period 1980–2014 (35 years)" [—Response]

L151: Which input was used for the weather generator?

[Response—] see [Page 8 Line 169-171]: "Two additional variables (wind speed and relative humidity) were estimated by the SWAT model's climate generator (Gassman et al., 2007), where the monthly weather database "WGEN_US_COOP_1980_2010" was used." [—Response]

L156: If the SWAT outlets are set at the same position as the location of HUC8 runoff data, then runoff is no intermediate variable.

[Response] The SWAT model generates runoff at the subbasin level. However, we need to compare runoff at the HUC8 level. See [Page 11 Line 228-234]: "As aforementioned, the USGS HUC8 runoff represents the local water yield (in units of mm), not the streamflow. There are 32 HUC8 units in the TRB, and the watershed was delineated into 55 subbasins for SWAT modeling. As a result, each HUC8 consists of 1–4 subbasins. For example, when we implemented calibration in terms of the HUC8-06010102, the parameters in four subbasins (1, 4, 5 and 6) in this HUC8 were calibrated. We calculated simulated HUC8 runoff as the area-weighted-average of runoff

from the SWAT subbasins within each HUC8". [—Response]

L200: When such long time series are available I recommend to use a larger fraction of it as warm up period. One year of warm up is very short.

[Response—] Thanks for the reviewer's constructive suggestion! With your suggestion, we have tried to use >1-year for warm-up in one HUC8 and did NOT find significant influence on our runoff calibration and validation performance, thus at current stage we decided not to re-do model calibration with >1-year warm-up period. [—Response]

L209 – 210: Why is it necessary to accumulate these runoffs? If the sub basins in the model are assigned properly then the runoff or water yield is explicitly given by the SWAT outputs.

[Response—] We added statements to further explain why it is necessary. See [Page 11 Line 228-234]: "As aforementioned, the USGS HUC8 runoff represents the local water yield (in units of mm), not the streamflow. There are 32 HUC8 units in the TRB, and the watershed was delineated into 55 subbasins for SWAT modeling. As a result, each HUC8 consists of 1–4 subbasins. For example, when we implemented calibration in terms of the HUC8-06010102, the parameters in four subbasins (1, 4, 5 and 6) in this HUC8 were calibrated. We calculated simulated HUC8 runoff as the area-weighted-average of runoff from the SWAT subbasins within each HUC8". [—Response]

L218 – 222: This statement still surprises me. There are data from 6000 Stations but none of these are adequate for use in validation?

[Response—] please see our [Response] to the major concern #1. [—Response]

L244 – 246: As stated above the defined thresholds are too low. Taking the absolute refuses one to identify positive or negative relationships.

[Response—] We did show positive or negative correlations in Fig. 6. We also re-defined the thresholds for correlations. please see our [Response] to the major concern #4. [—Response]

L272 – 277: For calibration the total range of the NSE values was given. For validation only the CI is shown. It is logical that some sub basins perform much worse during validation. Refusing to show this information gives however a wrong impression.

[Response—] Sorry for the missing information. We added the NSE range for runoff validation. See [Page 14 Line 313-315]: "The model also performed well over the validation period, although NSE (ranged −0.10–0.83) was lower than that during the calibration period. The median NSE was 0.72 with 50% CI of 0.57–0.77;" [—Response]

L294: I would not call it a great improvement when a model performs just as good as the mean value.

[Response—] We deleted the word "greatly" in this sentence. The NSE values for calibration of TP and TN were 0.44 and 0.38, respectively. The NSE for calibration of sediment is 0.06, as the reviewer pointed out, where the model performs just as good as the mean value, but the model performance is still improved compared to the uncalibrated model with a large negative NSE (−100). [—Response]

L321 -325: Why are SPARROW TP estimates highly correlated to SWAT TN estimates? These originate usually from different processes. This finding requires further analysis and discussion.

[Response—] We are sorry to have presented this inappropriate information. We deleted the statement "and (SPARROW-TP) highly correlated with SWAT-simulated TN (r = 0.84, p-value < 0.001)". SPARROW-TP was correlated with SPARROW-TN, and SWAT-TN was correlated with SPARROW-TN, which resulted in a significant correlation coefficient between SWAT-TN and SPARROW-TP, however, they might not be mechanistically correlated. Actually, different from the SPARROW estimates, SWAT-simulated spatial distributions of TN and TP were not correlated with each other. Please see our response to major concern #4 for detail. [—Response]